# Decepticons: Corrupted Transformers Breach Privacy in Federated Learning for Language Models

**Liam Fowl**[*], **Jonas Geiping**[*]
University of Maryland
{lfowl, jgeiping}@umd.edu

**Steven Reich**
University of Maryland

**Yuxin Wen**
University of Maryland

**Wojtek Czaja**
University of Maryland

**Micah Goldblum**
New York University

**Tom Goldstein**
University of Maryland
tomg@umd.edu

## Abstract

Privacy is a central tenet of Federated learning (FL), in which a central server trains models without centralizing user data. However, gradient updates used in FL can leak user information. While the most industrial uses of FL are for text applications (e.g. keystroke prediction), the majority of attacks on user privacy in FL have focused on simple image classifiers and threat models that assume honest execution of the FL protocol from the server. We propose a novel attack that reveals private user text by deploying malicious parameter vectors, and which succeeds even with mini-batches, multiple users, and long sequences. Unlike previous attacks on FL, the attack exploits characteristics of both the Transformer architecture and the token embedding, separately extracting tokens and positional embeddings to retrieve high-fidelity text. We argue that the threat model of malicious server states is highly relevant from a user-centric perspective, and show that in this scenario, text applications using transformer models are much more vulnerable than previously thought.

## 1 Introduction

Federated learning (FL) has recently emerged as a central paradigm for decentralized training. Where previously, training data had to be collected and accumulated on a central server, the data can now be kept locally and only model updates, such as parameter gradients, are shared and aggregated by a central party. The central tenet of federated learning is that these protocols enable privacy for users (McMahan & Ramage, 2017; Google Research, 2019). This is appealing to industrial interests, as user data can be leveraged to train machine learning models without user concerns for privacy, app permissions or privacy regulations, such as GDPR (Veale et al., 2018; Truong et al., 2021). However, in reality, these federated learning protocols walk a tightrope between actual privacy and the appearance of privacy. Attacks that invert model updates sent by users can recover private information in several scenarios Phong et al. (2017); Wang et al. (2018) if no measures are taken to safe-guard user privacy. Optimization-based inversion attacks have demonstrated the vulnerability of image data when only a few datapoints are used to calculate updates (Zhu et al., 2019; Geiping et al., 2020; Yin et al., 2021). To stymie these attacks, user data can be aggregated securely before being sent to the server as in Bonawitz et al. (2017), but this incurs additional communication overhead, and as such requires an estimation of the threat posed by inversion attacks against specific levels of aggregation, model architecture, and setting.

Most of the work on gradient inversion attacks so far has focused on image classification problems. Conversely, the most successful industrial applications of federated learning have been in language tasks. There, federated learning is not just a promising idea, it has been deployed to consumers in production, for example to improve keystroke prediction (Hard et al., 2019; Ramaswamy et al., 2019) and settings search on the Google Pixel (Bonawitz et al., 2019). However, attacks in this area have so

---

[*]Authors contributed equally. Order chosen randomly.

|  | Batch Size = 1 | Batch Size = 8 | Batch Size = 16 |
|---|---|---|---|
| Length 128 | Ancient Egyptian deities are the gods and goddesses worshipped Egypt ancient constitu. The beliefs and rituals myths these gods | Ancient Egyptian deities are the gods and goddesses worshipped in ancient Egypt. The beliefs view rituals surrounding these gods | Ancient Egyptian deities are the gods and goddesses worshipped in ancient Egypt. The beliefs view rituals surrounding these continue |
| Length 512 | Ancient Egyptian well are the gods and goddesses worshipped in ancient Egypt ◆ The beliefs whereas ritualsies these gods formed | Ancient Egyptian deities are the gods and goddesses worshipped in ancient vague. " beliefs and. tried these gods | Ancient Egyptian deities are the gods and goddess hours thoughts in ancient final conception divine beliefs and rituals and these |

**Figure 1:** An example reconstruction from a small GPT-2 model using a *Decepticon* attack, showing the first 20 tokens reconstructed from a randomly selected user for different combinations of sequence length and batch size on a *challenging* text fragment. Highlighted text represents *exact* matches for position and token.

far succeeded only on limited examples of sequences with *few* ($< 25$) tokens (Deng et al., 2021; Zhu et al., 2019; Dimitrov et al., 2022), even for massive models such as BERT (with worse recovery for smaller models). This leaves the impression that these models are already hard to invert, and limited aggregation is already sufficient to protect user privacy, without the necessity to employ stronger defenses such as local or distributed differential privacy (Dwork & Roth, 2013).

In this work, we revisit the privacy of transformer models. We focus on the realistic threat model where the server-side behavior is untrusted by the user, and show that a malicious update sent by the server can completely corrupt the behavior of user-side models, coercing them to spill significant amounts of user data. The server can then collect the original words and sentences entered by the user with straightforward statistical evaluations and assignment problems. We show for the first time that recovery of all tokens and most of their absolute positions is feasible even on the order of *several thousand* tokens and even when applied to small models only 10% the size of BERT discussed for FL use in Wang et al. (2021). Furthermore, instead of previous work which only discuss attacks for updates aggregated over few users, this attack breaks privacy even when updates are aggregated over more than 100 users. We hope that these observations can contribute to re-evaluation of privacy risks in FL applications for language.

## 2 MOTIVATION AND THREAT MODEL

At first glance, gradients from Transformer architectures might not appear to leak significant amounts of user data. Both the attention mechanisms and the linear components learn operations that act individually on tokens, so that their gradients are naturally averaged over the entire length of the sequence (e.g. 512 tokens). Despite most architectures featuring large linear layers, the mixing of information reduces the utility of their content to an attacker. In fact, the only operation that "sees" the entire sequence, the scaled dot product attention, is non-learned and does not leak separate gradients for each entry in the sequence. If one were to draw intuition from vision-based attacks, gradients whose components are averaged over 512 images are impossible to invert even for state-of-the-art attacks (Yin et al., 2021).

On the other hand, recovering text appears much more constrained than recovering images. The attacker knows from the beginning that only tokens that exist in the vocabulary are possible solutions and it is only necessary to find their location from a limited list of known positions and identify such tokens to reconstruct the input sequence perfectly.

**Previous Attacks** Previous gradient inversion attacks in FL have been described for text in Deng et al. (2021); Zhu et al. (2019); Dimitrov et al. (2022) and have focused on optimization-based reconstruction in the *honest-but-curious* server model. Recently, Gupta et al. (2022) use a language model prior to improve attack success via beam-search that is guided by the user gradient. Our work is further related to investigations about the unintended memorization abilities of fully trained, but

benign, models as described in Carlini et al. (2021); Inan et al. (2021); Brown et al. (2021), which can extract up to $1\%$ of training data in some cases (Carlini et al., 2022). In contrast, we investigate attacks which *directly* attempt to retrieve a large fraction of the training data utilized in a single model update. This is the worst-case for a privacy leak from successive benign model snapshots discussed in Zanella-Béguelin et al. (2020).

**Threat Model**    We consider the threat model of an untrusted server that is interested in recovering private user data from user updates in FL. Both user and server are bound by secure implementations to follow a known federated learning protocol, and the model architecture is compiled into a fixed state and verified by the same implementation to be a standard Transformer architecture. The user downloads a single malicious server state and returns their model updates according to protocol. The server then recovers user data from this update.

This threat model is the most natural setting from a user-centric perspective. Stronger threat models would, for example, allow the server to execute arbitrary code on user devices, but such threats are solvable by software solutions such as secure sandboxing (Frey, 2021). Still other work investigates malicious model architectures (Fowl et al., 2021; Boenisch et al., 2021), but such malicious modifications have to be present at the conception of the machine learning application, before the architecture is fixed and compiled for production (Bonawitz et al., 2019), making those attacks most feasible for actors in the "analyst" role (Kairouz et al., 2021b). Ideally, user and server communicate through a verified protocol implementation that only allows pre-defined and vetted model architectures.

However, none of these precautions stop an *adversarial server* from sending malicious updates (Wang et al., 2021). Such attacks are naturally ephemeral - the server can send benign server states nearly all of the time to all users, then switch to a malicious update for single round and group of users (or single secure aggregator), collect user information, and return to benign updates immediately after. The malicious update is quickly overwritten and not detectable after the fact. Such an attack can be launched by any actor with temporary access to the server states sent to users, including, aside from the server owner, temporary breaches of server infrastructure, MITM attacks or single disgruntled employees. Overall we argue that server states sent to users, especially for very expressive models such as transformers, should be treated by user applications as analogue to *untrusted code* that operates on user data and should be handled with the appropriate care.

## 3    MALICIOUS PARAMETERS FOR DATA EXTRACTION

Our attack leverages several interacting strategies that we present in the sub-sections below.

For simplicity, we assume in this section that the FL protocol is fedSGD, i.e. model update returned by the users is the average gradient of the model parameters sent by the server computed over all local user data. If the server is malicious, then this protocol can either be selected directly or another protocol such as federated averaging (fedAVG), which includes multiple local update steps (McMahan et al., 2017), could be modified maliciously to run with either only a single local step or a large enough batch size. Both variants effectively convert the protocol to fedSGD. The local user data is tokenized text fed into the transformer model, although the attack we describe does not make use of text priors and can read out an arbitrary stream of tokens.

We separate the exposition into two parts. First, we discuss how the server constructs a malicious parameter vector for the target architecture that is sent to users. Second, we then discuss how the server can reconstruct private information from this update.

We first discuss parameter modifications for a single sequence of tokens, and then move to modifications that allow recovery of multiple sequences. For the following segment, we assume that the targeted model is based on a standard transformer architecture, containing first an embedding component that, for each token, adds token embedding and positional embedding, and second, a series of transformer layers, each containing an attention block and a feed-forward block.

**Modifications to Recover Token Ids and Positions:**    In principle, the goal of the attacker is simple: If the attacker can reconstruct the inputs to the first transformer layer, which are token embeddings plus positional embeddings, then a simple matching problem can identify the token id and position id by correlating each input to the known token and position embedding matrices, see **??**. However,

what the user returns is not a token but a gradient, representing changes to the model relative to the user's data. This gradient is also an average over all tokens in the sequence. As a first step to improve attack success, the attacker can *disable all attention layers* (setting their output transformation to 0) and mostly *disable outputs of each feed-forward block*. The outputs of each feed-forward layer cannot be entirely disabled (otherwise all gradients would be zero), so we set all outputs except the last entry to zero. A transformer modified in this way does not mix token information and the inputs to each layer are the same (except for the last entry in the hidden dimension) and equal to token embeddings plus positional embeddings.

**Unmixing Gradient Signals:** Let's investigate the resulting gradient of the first linear layer in each feed-forward block. We define the input embeddings that the attacker intends to recover as $\{u_k\}_{k=1}^S$ coming from a sequence of text containing $S$ elements. The operation of the first linear layer in each ($i^{th}$) feed-forward block is $y_k = W_i u_k + b_i$. The gradient averaged over all tokens in this sequence is:

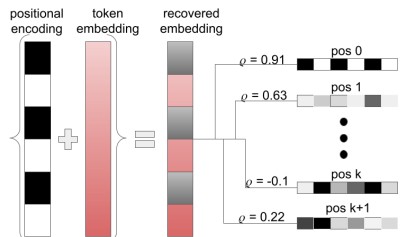

**Figure 2:** Recovering each token and its positions from the inputs to the first transformer layer. We find the correlation matrix between recovered input embeddings to known positions and tokens, and solve a sum assignment problem to determine which position is most likely for each input.

$$\sum_{k=1}^S \nabla_{W_i^{(j)}} \mathcal{L}_k = \sum_{k=1}^S \left( \frac{\partial \mathcal{L}_k}{\partial y_k^{(j)}} \cdot \nabla_{W_i^{(j)}} y_k^{(j)} \right), \tag{1}$$

$$\sum_{k=1}^S \frac{\partial \mathcal{L}_k}{\partial b_i^{(j)}} = \sum_{k=1}^S \frac{\partial \mathcal{L}_k}{\partial y_k^{(j)}} \cdot \frac{\partial y_k^{(j)}}{\partial b_i}, \tag{2}$$

For each of the $j$ entries, rows of the bias and weight, respectively. This can be simplified further to: $\sum_{k=1}^S \frac{\partial \mathcal{L}_k}{\partial y_k^{(j)}} \cdot u_k$ for weight and $\sum_{k=1}^S \frac{\partial \mathcal{L}_k}{\partial y_k^{(j)}}$ for bias (Geiping et al., 2020). Even after this reduction, it seems quite difficult to recover $u_k$ from these sums. Yet, a malicious attacker can modify the weights of $W_i$ and of the subsequent matrix to allow for a closed-form solution.

To this end, we leverage recent strategies for separating gradient signals (Fowl et al., 2021). In essence, this strategy encodes a measurement function into each weight row $W_i^{(j)}$ of a linear layer $i$ and an increasing offset into each entry of the bias vectors $b_i$. For linear layers that are followed by activations which cut off smaller input values (such as ReLU or GELU), the gradient of the weights in this linear layer will then encode a cumulative sum of the values of all inputs up to some offset value. This structures the gradients observed in Equation (1) and original inputs $u_k$ can then be recovered by subtraction of adjacent rows. This is visualized in Figure 3.

To go further into detail into this modification, the rows of all weight matrices $W_i$ are replaced by a measurement vector $m$ and the biases over all layers are modified to be sequentially ascending. As a result, all linear layers now compute $y_k^{(j)} = \langle m, u_k \rangle + b^{(j)}$. And $y_k$ is ordered so that $y_k^{(j)} \geq y_k^{(j')}, \forall j < j'$. The subsequent ReLU unit will threshold $y_k^{(j)}$, so that $y_k^{(j)} = 0$ directly implies $\langle m, u_k \rangle < -b^{(j)}$. This ordering can now be exploited by subtracting the $(j+1)^{th}$ entry in the gradient from the $j^{th}$ entry to recover:

$$\sum_{k=1}^S \nabla_{W_i^{(j)}} \mathcal{L}_k - \sum_{k=1}^S \nabla_{W_i^{(j+1)}} \mathcal{L}_k = \lambda_i^{(j)} \cdot u_k, \qquad \sum_{k=1}^S \frac{\partial \mathcal{L}_k}{\partial b^{(j)}} - \sum_{k=1}^S \frac{\partial \mathcal{L}_k}{\partial b^{(j+1)}} = \lambda_i^{(j)} \tag{3}$$

for scalars $\lambda_i^{(j)}$. Assuming the number of embeddings that fulfill this condition is either one or zero, we recover the input vector $u_k$ simply by dividing the term of the left of Equation (3) by the term on the right. All embeddings that *collide* and fall into the same interval are not easily recoverable.

We minimize the probability of a collision by first sampling a Gaussian vector $m \sim \mathcal{N}(\vec{0}, \mathbb{1}_d)$ where $d = d_{model}$. We then set the biases to $b_i = [c_{i \cdot k}, \ldots, c_{(i+1) \cdot k - 1}]$, where $k$ is the width of each linear layer and $c_l = -\Phi^{-1}(\frac{l}{M})$, where $\Phi^{-1}$ is the inverse of a Gaussian CDF and $M$ is the sum over the widths of all included Transformer blocks. This uniformly distributes the values of $\langle m, u_k \rangle$ over

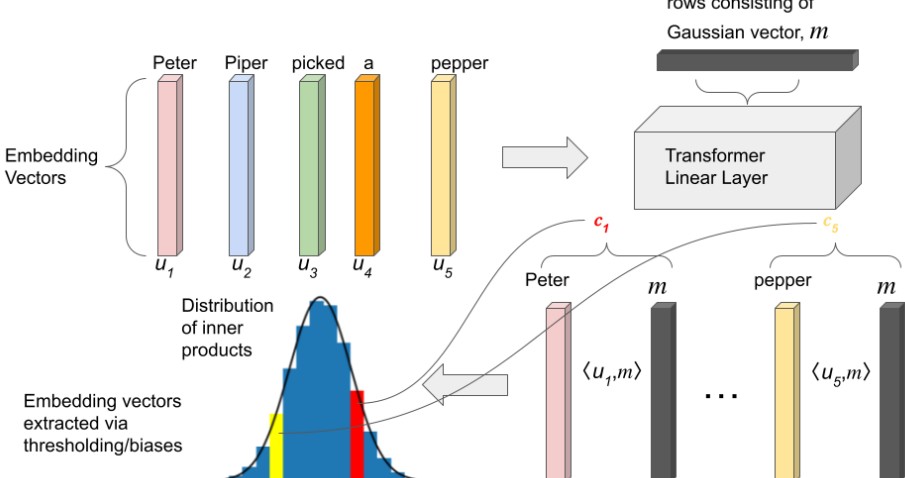

**Figure 3:** How embeddings are extracted from linear layers in the transformer blocks. We initialize the rows of the first linear layer in each Transformer block to a randomly sampled Gaussian vector (all rows being the same). The bottom portion of the figure depicts the internals of a forward pass through this FC layer. When an incoming embedding vector $u_k$ enters the FC layer, the inner products $\langle m, u_k \rangle$ fall into a distribution which the attacker partitions into bins. If $u_k$ is the only token whose inner product lies in a given bin, the ReLU activation will encode information about this single token, and (3) can be used to directly recover the token.

the bins created by the bias vector. If the width of the first linear layer in each block is larger than the number of tokens, collisions will be unlikely. We can estimate the rough mean and variance of $\{\langle m, u_k \rangle\}$ from either a small volume of public text or a random stream of tokens, see Appendix D.

**Disambiguating Multiple Sequences** Recovering multiple sequences – either from user data comprising multiple separate sentences of tokens, or aggregates of multiple users – presents the most difficult task for the attacker. Naively using the strategy described so far can only recover a partially ordered set of tokens. For example, if a model update consists of five user sequences, then the attacker recovers five tokens at position 0, five tokens at position 1, and so on, because positional embeddings repeat in each sequence. Re-grouping these tokens into salient sequences quickly becomes intractable as the number of possible groupings grows as $n^l$ for $n$ sequences of length $l$. Further complicating matters for the attacker is that no learned parameters (and thus no parameters returning gradients) operate on the entire length of a sequence in the Transformer model. In fact, the only interaction between embeddings from the same sequence comes in the scaled dot product attention mechanism.

Thus, if any malicious modification can be made to encode sequence information into embeddings, it must utilize the attention mechanism of Transformers. We describe a mechanism by which an attacker can disambiguate user sequences, even when model updates are aggregated over a large number of separate sequences. At a high level, we introduce a strategy that encodes unique information about each sequence into a predefined location in the embedding of each token in the sequence. Then, we can read out a unique quantity for each sequence and thus assign tokens to separate sequences. This behavior is illustrated in Figure 4.

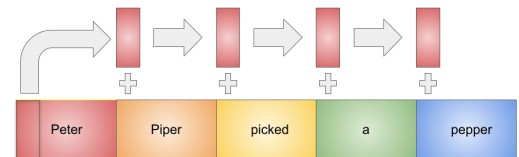

**Figure 4:** A high-level schematic of MHA manipulation. The MHA block attends to the first word identically for every input sequence, encoding a part of the first token for each embedding in the entire sequence.

Let $W_Q, W_K, W_V$ represent the query, key, and value weight matrices respectively, and let $b_Q, b_K, b_V$ represent their biases. For simplicity of presentation, we explain this attack on a single head, but it is easily adapted to multi-head attention. We first set the $W_K$ matrix to the identity ($\mathbb{1}_{d_{model}}$), and $b_K = \vec{0}$. This leaves incoming embeddings unaltered. Then, we set $W_Q = \vec{0}$, and $b_Q = \gamma \cdot \vec{p_0}$ where $\vec{p_0}$ is the first positional encoding. Here we choose the first position vector for simplicity, but there are many potential choices for the identifying vector. This query matrix then transforms each embedding identically to be a scaled version of the first positional encoding. We then set $W_V = \mathbb{1}_{d'}$

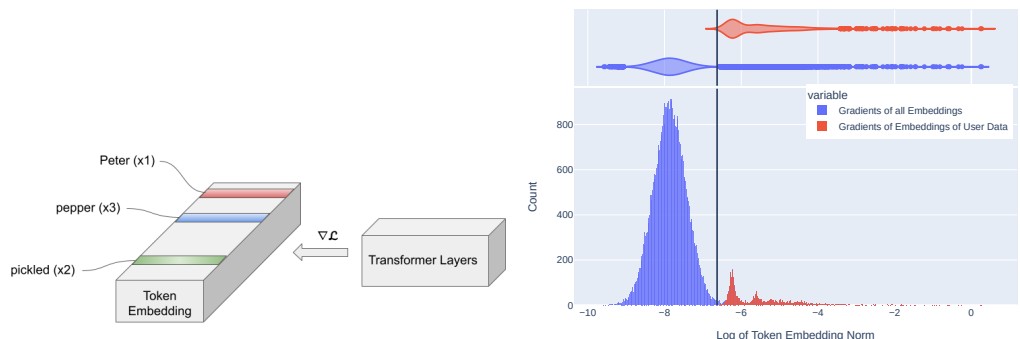

**Figure 5: Left:** A high-level schematic of token leaking. The token embedding layer leaks tokens and frequencies solely through its sparse gradient entries. **Right:** Distribution and Histogram of log of norms of all token embedding gradients for GPT-2 for $13824$ tokens. In this case, gradient entries are non-sparse due to the tied encoder-decoder structure of GPT, but the embeddings of true user data (red) are clearly separable from the mass of all embeddings (blue) by a cutoff at 1.5 standard deviations (marked in black).

to be a partial identity (identity in the first $d'$ entries where $d' \leq d$ is a hyperparameter that the server controls). Finally, we set $b_v = \vec{0}$.

Now, we investigate how these changes transform an incoming sequence of embeddings. Let $\{u_k\}_{i=0}^{l-1}$ be embeddings for a sequence of length $l$ that enters the attention mechanism. $u_0$ is the embedding corresponding to the first token in the sequence, so $u_0$ is made up of the token embedding for the first token in the sequence, and the first positional encoding. $W_K$ produces keys $K$ that exactly correspond to the incoming embeddings, however, $W_Q, b_Q$ collapses the embeddings to produce $Q$, consisting of $l$ identical copies of a single vector, the first positional encoding. Then, when the attention weights are calculated as:

$$\mathrm{softmax}\left(\frac{QK^T}{\sqrt{d_k}}\right),$$

the attacker finds that the first embedding dominates the attention weights for all the embeddings, as the query vectors all correlate with the first embedding the most. In fact, the $\gamma$ parameter can effectively turn the attention weights to a delta function on the first position. Finally, when the attention weights are used to combine the values $V$ with the embeddings, by construction, a part of the embedding for the first word in the sequence is identically added to each other word in that sequence. So the embeddings are transformed as $\{x_i\}_{i=0}^{l-1} \rightarrow \{x_i + x_{0,d'}\}_{i=0}^{l-1}$ where $x_{0,d'}$ is a vector where the first $d'$ entries are the first $d'$ entries of $x_0$, and the other $d_{model} - d'$ entries are identically 0.

If the attacker chooses to perform this modification on the first Transformer block, this means that embeddings $u_k$ that the attacker recovers from each of the linear layers *now also contain unique information about the first token of the sequence from which they came*. The attacker can then calculate correlations between first-position embeddings and later-position embeddings and group each embedding into a sequence with other embeddings.

### 3.1 EXTRACTING USER DATA

Now that we have introduced the main mechanisms by which an attacker prepares a malicious parameter vector, we summarize how to extract user data. The attack begins after a user or aggregate group of users has computed their gradient using the corrupted parameters. Then, the server retrieves their update and begins the inversion procedure. We summarize the entire attack in Algorithm 1.

**Getting a Bag-of-Words Containing All Tokens** Even without malicious parameters, an attacker can retrieve the bag-of-words (unordered list of tokens) of the user from the gradient of the token embedding. Melis et al. (2019) identified that the embedding layer gradient is only non-zero at locations corresponding to tokens that were used for training, as visualized in the left plot of Figure 5. Furthermore, the frequency of all words can be extracted by analyzing the bias gradient of the embedding or decoding (last linear) layer, as the magnitude of the update of each row of a random embedding matrix is proportional to the frequency of word usage, allowing for a greedy estimation by adapting the strategy of Wainakh et al. (2021) which was proposed for simpler classification tasks.

---

**Algorithm 1** Decepticon Data Readout Overview

---

1: **Input:** Sequence Length $S$, number of expected sequences $N$. Gradients $\nabla W_i, \nabla b_i$ of the weight and bias of the first linear layer in each FFN block. Positional embeddings $P$, token embeddings $T$.
2: $t_v \leftarrow$ Token Embeddings of estimated bag-of-words of leaked tokens or all tokens
3: $p_k \leftarrow$ Known positional embeddings
4: $u_h \leftarrow \frac{\nabla_{W_{ij}} - \nabla_{W_{i,j+1}}}{\nabla_{b_j^i} - \nabla_{b_{j+1}^i}}$ for linear layers $i = 1, ..L$ and rows $j = 1, ..., r$.
5: $L_{\text{batch}} \leftarrow$ Cluster index for each $u_h$ clustering the first $d'$ entries of each embedding.
6: **for** n in $0...N$ **do**
7:      $u_{hn} \leftarrow$ Entries of cluster $n$ in $L_{\text{batch}}$
8:      $u_k^n \leftarrow \text{Match}(P_k, u_{hn})$ on all entries $d > d'$
9: **end for**
10: $u_{nk} \leftarrow$ concatenate $\{u_k^n\}_{n=1}^N$
11: $t_{nk}^{\text{final tokens}} \leftarrow$ Indices of $\text{Match}(u_{nk} - p_k, t_v)$ on all entries $d > d'$

---

We use this finding as a first step to restrict the number of possible tokens that should be matched. More details can be found in Appendix C.

**Recovering Input Embeddings:** As a first step, we recover all input embeddings $u_h$ using the divided difference method described in Equation (3) for all rows where $\nabla_{b_j^i} - \nabla_{b_{j+1}^i}$ is non-zero, i.e. at least one embedding exist with $-b_j < \langle m, u_k \rangle < -b_{j+1}$. This results in a list of vectors $u_h$ with $h < SN$, for maximal sequence length $S$ and maximal number of expected sequences $N$, that have to be matched to separate sequences, positions and tokens.

**Recovering Sequence Clusters:** The first $d'$ entries of each embedding encode the sequence identity as described in Section 3. The server can hence group the embeddings $u_h$ uniquely into their sequences by a clustering algorithm. We utilize constrained K-Means as described in (Bradley et al., 2000), as the maximal sequence length $S$ and hence cluster size is known. This results in a label for each vector $u_h$, sorting them into separate sequences.

**Assigning Positions in each Sequence:** Now, the matching problem has been reduced to separately matching each sequence, based on all entries $d > d'$. For a given sequence with index $n$ first match the embedding $u_{hn}$ to the available positional embeddings $P_k$. To do so, we construct the matrix of correlations between all $u_{hn}$ and $p_k$ and solve a rectangular linear sum assignment to find the optimal position $p_k$ for each $u_{hn}$. For the assignment problem, we use the solver proposed in Crouse (2016) which is a modification of the shortest augmenting path strategy originally proposed in Jonker & Volgenant (1987). Due to collisions, some positions will possibly not be filled after this initial matching step. We thus iteratively fill up un-matched positions with the best-matching embeddings from $u_{hn}$, even if they have been used before in other positions, until each position is assigned a match.

**Converting Embeddings to Tokens:** Finally, we match all $u_{nk}$ to all token embeddings $t_v$ in either our bag of possible tokens or all tokens in the vocabulary. For this, we first subtract or decorrelate the positional encoding $p_k$ from $u_{nk}$ to remove interference, and then again solve a linear sum assignment problem as described. The indices of the solution to this assignment problem will finally be the token ids $t_{nk}$ for each position $k$ in sequence $n$.

In general, this process works very well to recover batches containing multiple long sequences. In the limit of many tokens, the quality of recovery eventually degrades as collisions become more common, making the identification of positions and token id less certain. In the regime of a massive number of tokens, only a subset of positions are accurately recovered, and the remaining tokens are inaccurate, appearing in wrong positions. This process is random, and even for a massive number of tokens there are often some sequences that can be recovered very well.

## 4 EMPIRICAL EVALUATION OF THE ATTACK

We evaluate the proposed attack using a range of Transformers. Our main focus is the application to next-word prediction as emphasized in practical use-cases (Hard et al., 2019; Ramaswamy et al., 2019). We consider three architectures of differing sizes: the small 3-layer Transformer discussed as a template for federated learning scenarios in Wang et al. (2021) (11M parameters), the BERT-Base

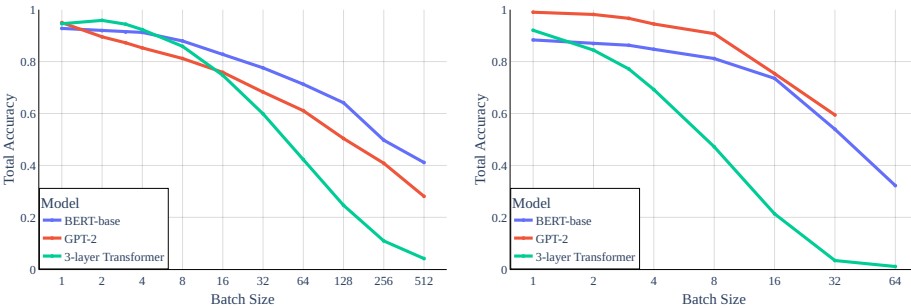

**Figure 6: Total Average Accuracy**. Total accuracy (i.e. percentage of tokens recovered correctly in their correct position) for all considered and varying batch sizes for sequence length 32 from `wikitext`(left) and sequence length 512 from `stackoverflow` (right).

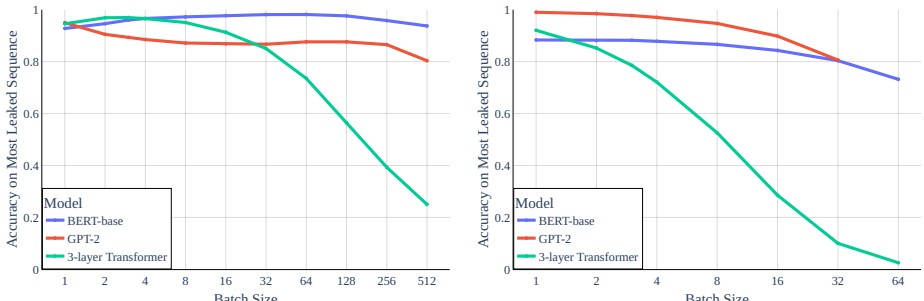

**Figure 7: Most-vulnerable Accuracy**. Accuracy on the most-leaked sentence for sequence length 32 from `wikitext`(left) and sequence length 512 from `stackoverflow` (right). Note the size difference between BERT/GPT and the small transformer depicted here.

model (Devlin et al., 2019) attacked in previous work (Deng et al., 2021; Zhu et al., 2019) (110M parameters), and the smallest GPT-2 variation (Radford et al., 2019) (124M parameters). We train the small Transformer and GPT-2 as causal language models and BERT as a masked language model. Predictably, the attack performance degrades most quickly for the 3-layer Transformer as this has the fewest parameters and thus the least capacity to uniquely capture a large number of user embeddings.

We test this attack on several datasets, `wikitext` (Merity et al., 2016) `shakespeare` (Caldas et al., 2019) and `stackoverflow` (Wang et al., 2021). For `wikitext`, We partition into separate "users" by article, while for the other datasets user partitions are given. We tokenize using the GPT-2 (BPE) tokenizer for the small Transformer and GPT-2, and the original BERT (WordPiece) tokenizer for BERT. We always report average scores over the first 100 users, which are have enough data to fill batch size × sequence length tokens, skipping users with less data who would be more vulnerable. We focus on fedSGD, i.e. single gradient updates from all users, but note the option of a malicious server to convert another protocol to fedSGD discussed in Section 3.

We evaluate using *total* average accuracy, BLEU (Papineni et al., 2002) and ROUGE-L (Lin, 2004). Our total accuracy is stricter than the token (i.e., bag-of-words) accuracy described previously Deng et al. (2021); we only count success if both token id *and* position are correct. Figure 1 shows partial reconstructions for a randomly selected, challenging sequence as batch size and sequence length increase. We find that even for more difficult scenarios with more tokens, a vast majority of tokens are recovered, with a majority in their *exact* correct position.

**Average Accuracy:** Our multi-head attention strategy allows an attacker to reconstruct larger batches of sequences, as seen in Figure 6 (We include single sequences in Appendix E.2). This applies to an update from a single user with more than one sequence, and also multiple users aggregating model updates. To the best of our knowledge, this is the first approach to explicitly attack multi-user updates with *strong average* accuracy. We find that for a sequence length of 32, the proposed attack can recover almost all tokens used in the user updates, and $> 50\%$ of tokens at their *exact* position for $> 100$ users. Even for a sequence length of 512, the attack remains effective for large batches.

**Most-vulnerable Accuracy:** We further chart the accuracy on the most-recovered sequence in Figure 7, finding that even at larger aggregation rates, some sequences remain vulnerable to attack and are almost perfectly recovered by an attacker.

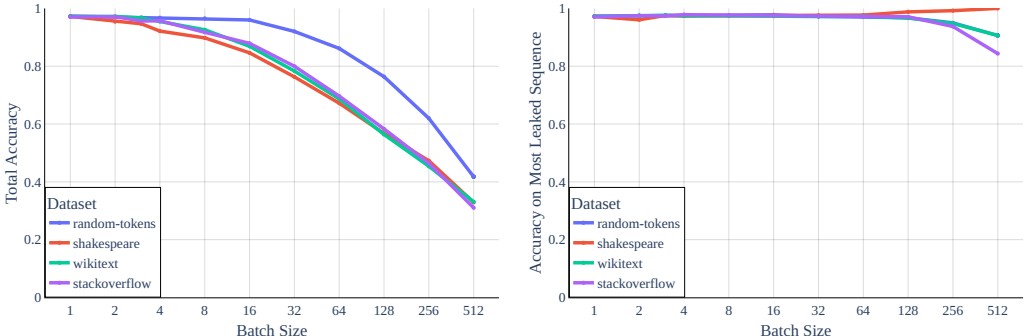

**Figure 8: Comparing attacks on several data sources**. **Left:** Total accuracy (i.e. percentage of tokens recovered correctly in their correct position) for GPT-2 and varying batch sizes with sequence length 32. **Right:** Accuracy on the most-leaked sentence for all data sources.

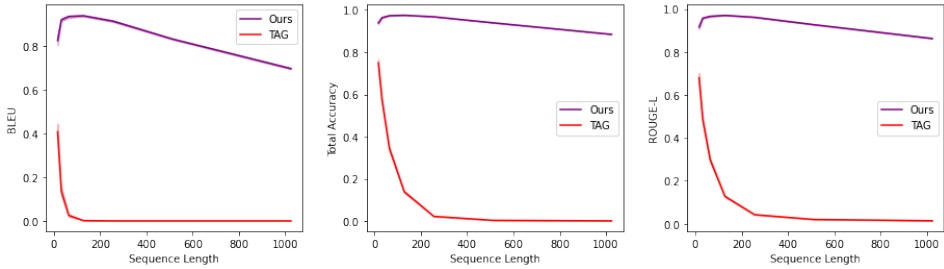

**Figure 9:** Comparison between the malicious server threat model with our attack and TAG (honest-but-curious) for the 3-layer transformer described in Wang et al. (2021), variable seq. length (batch size 1) on `wikitext`.

**Comparison to other threat models.** Several works approached the topic of reconstructing text from FL updates, under a "honest-but-curious" threat model where benign parameters are sent to the user, and the FL protocol is performed normally. We compare by fixing an architecture – in this case the Transformer-3 model described earlier. We then consider the setting with batch size 1 and evaluate our proposed method against the TAG attack Deng et al. (2021) which improves upon previous results in Zhu et al. (2019). We experiment for varying sequence lengths in Figure 9. The performance of TAG very quickly degrades for sequences of even moderate length. For a single sequence, our proposed attack maintains high total accuracy for sequences exceeding 1000 tokens, whereas the total accuracy for TAG soon approaches $0\%$. Overall, we find that the severity of the threat posed by our attack is orders of magnitude greater than in the "honest-but-curious" threat model, and argue for the re-evaluation of the amount of privacy leaked by FL applications using transformers.

**Accuracy on other Datasets:** The attack is largely invariant to the source of user text, as detailed in Figure 8, where diverse sources of data are almost equally vulnerable. Shakespeare data is most vulnerable due to the use of short and simple sentences which are consistently well-recovered. Random text is noticeably more vulnerable, which is relevant from a security perspective, as an attacker often be interested in text fragments that are not usual text, e.g. passwords or social security numbers.

**Attack Confidence** In contrast to existing attacks, the attacker can verify recovered tokens. After recovering positions and tokens for each breach $u_h$, one can cross-check whether the assigned position $p_k$ and token embedding $t_v$ perfectly explain $u_h$. Tokens that fulfill this condition are perfectly recovered and allow the attacker to certify which parts of their recovered text accurately reflect user data.

## 5 CONCLUSIONS

In this work we re-evaluate the threat posed by attacks against privacy for FL for Transformer-based language models. We argue that from a user-perspective, the most natural threat model is not to trust the server state. We show that a malicious server can send updates that encode large amounts of private user data into the update sent by users. This attack and threat model significantly lower the bar for possible attacks, recovering up to several thousand tokens of user data. Our results underline the necessity of adoption of strong privacy guarantees for users in federated learning.

ETHICS STATEMENT AND MITIGATIONS

Several known case studies of FL models deployed in production in Hard et al. (2019); Hartmann (2021) rely only on aggregation to preserve privacy in FL for text. However, the attack we describe in this work shows that aggregation levels considered safe based on previous work may not be enough to protect user privacy in several use cases: The setting deployed to production in Hard et al. (2019) runs FL on users updates with $400$ sentences with $4$ words per average message sent to the server without further aggregation, well within the range of the investigated attack if trained with a transformer model. We thus briefly discuss other mitigation strategies for attacks like this. We roughly group them into two categories: parameter inspection and differential privacy based defenses.

Parameter inspection and verification of the server state is currently not implemented in any major industrial FL framework (Paulik et al., 2021; Dimitriadis et al., 2022; Li et al., 2019; Bonawitz et al., 2019), but after the publication of this attack, a rule could be designed to mitigate it (which we would encourage!). However, we caution that the design space for attacks as described here seems too large to be defended by inspecting parameters for a list of known attacks.

Based on these concerns, general differential privacy (DP) based defenses continue to be the more promising route. Local or distributed differential privacy, controlled directly by the user and added directly on-device (instead of at the server level as in McMahan et al. (2018)) allows for general protection for users without the need to trust the update sent by the server (Kairouz et al., 2021a; Bhowmick et al., 2019). Local DP does come at a cost in utility, observed in theoretical (Kairouz et al., 2021b; Kasiviswanathan et al., 2011; Ullman, 2021) as well as practical investigations (Jayaraman & Evans, 2019) when a model is pretrained from scratch (Li et al., 2022; Noble et al., 2022). DP currently also comes at some cost in fairness, as model utility is reduced most for underrepresented groups (Bagdasaryan et al., 2019). Yet, as any attack on privacy will break down when faced with sufficient differential privacy, with this work we advocate for strong differential privacy on the user side, as incorporated for example into recent system designs in Paulik et al. (2021).

We include further discussion about mitigations in practice in Appendix G, including limited experimentation with gradient clipping/noising, as well as parameter inspection adaptations.

Attacks like the one presented in this paper present a privacy risk to users and their data. As such, the publication of the possibility of attacks like the one described in this work has potential negative consequences for the privacy of deployed user systems that do not apply mitigation strategies as described above. As FL becomes more integrated into privacy-critical applications, it is of paramount importance that the community is aware of worst-case threats to user privacy, so that mitigating measures can be taken. If left undefended, an attack like this could compromise large amounts of user data, including sensitive text messages and emails. Because previous research on privacy attacks against language models in FL has been relatively scarce, some might believe that simple defenses such as aggregation are sufficient to ensure user privacy. This work shows that this belief is unfounded and further measures must be taken. We hope that this attack raises awareness of the vulnerability of FL models.

REPRODUCIBILITY STATEMENT

We provide technical details in Appendix A, and additional pseudocode for mentioned algorithms in Appendix B and provide code in conjunction with this submission to reproduce all attacks and settings discussed in this work. Within our code submission we provide command-line interfaces and jupyter notebooks to evaluate the attacks and interactively vary parameters. All attacks discussed in this work can be cheaply reproduced using Laptop CPUs, as no extensive computations on GPUs are necessary to run the attack.

ACKNOWLEDGEMENTS

This work was supported by the Office of Naval Research (#N000142112557), the AFOSR MURI program, DARPA GARD (HR00112020007), the National Science Foundation (IIS-2212182), and Capital One Bank.

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

## A  TECHNICAL DETAILS

We implement all attacks in an extensible `PyTorch` framework (Paszke et al., 2017) for this type of attack which allows for a full reproduction of the attack and which we attach to this submission. We utilize `huggingface` models and data extensions for the text data (Wolf et al., 2020).The attack is separated into two parts. The first part is the malicious modification of the server parameters, which is triggered immediately before the server sends out their model update payload to users.

We implement the malicious parameter modifications as described in Sec. 3. We initialize from a random model initialization and reserve the first $6$ entries of the embedding layer for the sentence encoding for the Transformer-3 model and the first 32 entries for the larger BERT and GPT-2 models. The corresponding entries in the positional and token embeddings are reset to $0$. In the MHA modification, we choose a softmax skewing of $1e8$, although this value can be significantly lower in practice as well. We reserve the last entry of the embedding layer for gradient flow to each linear layer, so that the attack is able to utilize all layers as described. This entry is scaled by $\varepsilon = 1e - 6$ for the smaller Transformer-3 model and $\varepsilon = 1e - 8$ for the larger models, so that the layer normalization is not skewed by large values arising from the last embedding entry.

For the attack part, we retrieve the gradient update on the server and run all recovery computations in single float precision. We first run a token recovery as discussed in Sec 3.1, which we additionally summarize in Sec. 3.4 and then proceed with the attack following Algorithm 1. For the sentence labeling we cluster using constrained K-means as described in Bradley et al. (2000). For all assignment problems we utilize the linear sum assignment solver proposed in Crouse (2016) which is a modification of the shortest augmenting path strategy originally proposed in Jonker & Volgenant (1987).

During the quantitative evaluation, we trial 100 users each with a request for updates of size sequence length × batches. We skip all users which do *not* own enough data and do not pad user data with [PAD] tokens, which we think would skew results as it would include a large number of easy tokens to recover. All measurements are hence done on non-trivial sentences, which are concatenated to reach the desired sequence length. Each user's data is completely separated, representing different wikipedia articles as described in the main body. Overall, the attack is highly successful over a range of users even with very different article content.

To compute metrics, we first resort the recovered batches toward the ground truth order by assigning to ground truth batches based on total token accuracy per sentence. This leads to a potential underestimation of the true accuracy and ROUGE metrics of the recovery as sentence are potentially mismatched. We compute Rouge (Lin, 2004) scores based on the sorted batches and BLEU scores (Papineni et al., 2002) (with the default `huggingface` implementation) by giving all batches as references. We measure total accuracy as the number of tokens that are perfectly identified and in the correct position, assigning no partial credit for either token identity or position. This is notably in contrast to the naming of metrics in Deng et al. (2021) where accuracy refers to only overall token accuracy. We refer to that metric as "token accuracy", measuring only the overlap between reference and reconstruction sentence bag-of-words, but report only the total accuracy, given that token accuracy is already near-perfect after the attack on the bag-of-words described in Sec 3.1.

# B ALGORITHM DETAILS

We detail sub-algorithms as additional material in this section. Algorithm 2 and Algorithm 3 detail the token recovery for transformers with decoder bias and for transformer with a tied embedding. These roughly follow the principles of greedy label recovery strategy proposed in Wainakh et al. (2021) and we reproduce them here for completeness, incorporating additional considerations necessary for token retrieval.

---

**Algorithm 2** Token Recovery
(Decoder Bias)

1: **Input:** Decoder bias gradient $g_b$, embedding gradient $g_e$, sequence length $s$, number of sequences $n$.
2: $v_{\text{tokens}} \leftarrow$ all indices where $g_b < 0$
3: $v_e \leftarrow$ all indices where $g_e < 0$
4: $v_{\text{tokens}}$ append $v_{\text{tokens}} \setminus v_e$
5: $m_{\text{impact}} = \frac{1}{sn} \sum_{i \in v_{\text{tokens}}} g_{bi}$
6: $g_b[v_{\text{tokens}}] \leftarrow g_b[v_{\text{tokens}}] - m_{\text{impact}}$
7: **while** Length of $v_{\text{tokens}} < sn$ **do**
8: $\quad j = \arg\min_i g_{bi}$
9: $\quad g_{bj} \leftarrow g_{bj} - m_{\text{impact}}$
10: $\quad v_{\text{tokens}}$ append $j$
11: **end while**

---

**Algorithm 3** Token Recovery
(Tied encoder Embedding)

1: **Input:** Embedding weight gradient $g_e$, sequence length $s$, number of expected sequences $n$, cutoff factor $f$
2: $n_{ej} = ||g_{ej}||_2$ for all $j = 1, ...N$ embedding rows
3: $\mu, \sigma = \text{Mean}(\log n_e), \text{Std}(\log n_e)$
4: $c = \mu + f\sigma$
5: $v_{\text{tokens}} \leftarrow$ all indices where $n_e > c$
6: $m_{\text{impact}} = \frac{1}{sn} \sum_{i \in v_{\text{tokens}}} n_{ei}$
7: $n_e[v_{\text{tokens}}] \leftarrow n_e[v_{\text{tokens}}] - m_{\text{impact}}$
8: **while** Length of $v_{\text{tokens}} < sn$ **do**
9: $\quad j = \arg\min_{i \in v_{\text{tokens}}} n_{ei}$
10: $\quad n_{ej} \leftarrow n_{ej} - m_{\text{impact}}$
11: $\quad v_{\text{tokens}}$ append $j$
12: **end while**

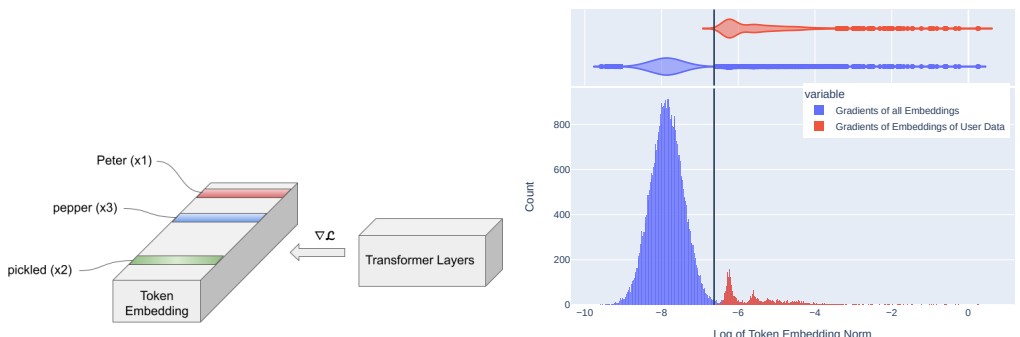

**Figure 10: Left:** A high-level schematic of a simple case of token leaking. The token embedding layer can leak tokens and frequency solely through its sparse gradient entries. **Right:** Distribution and Histogram of log of norms of all token embedding gradients for GPT-2 for 13824 tokens. In this case, gradient entries are non-sparse due to the tied encoder-decoder structure of GPT, but the embeddings of true user data (red) are clearly separable from the mass of all embeddings (blue) by a simple cutoff at 1.5 standard deviations (marked in black).

## C   TOKEN BAG-OF-WORDS RECOVERY

Even without any malicious parameter modifications, an attacker can immediately retrieve the bag-of-words (the unordered list of all tokens and their assorted frequencies) of the user data from the gradient of the token embedding. Melis et al. (2019) previously identified that unique tokens can be recovered due to the sparsity of rows in the token embedding gradient, as visualized in the left plot of Figure 5. But, perhaps surprisingly, we find that even the frequency of all words can be extracted. Depending on the model architecture, this frequency estimation can either be triggered by analyzing either the bias gradient of the decoding (last linear) layer, or the norm of the embedding matrix. In both cases, the magnitude of the update of each row of a random embedding matrix is proportional to the frequency of word usage, allowing for a greedy estimation by adapting the strategy of Wainakh et al. (2021) which was originally proposed to extract label information in classification tasks.

For models which contain a decoder bias parameter, this estimation directly leads to a $> 99\%$ accurate estimate of word frequencies. For models without decoder bias, such as GPT (Radford et al., 2019), the same strategy can again be employed based on the magnitude of embedding rows. We can further accurately predict token frequency even for models which tie their embedding and decoding weights, and as such do not feature naturally sparse embedding gradients. However, examining the distributions of embedding row norms in log-space for tokens contained in the other updates and all tokens, as visualized in the right plot of Figure 5, we find that unused tokens can be removed by a simple thresholding. The resulting token frequency estimation is inexact due to the cut-off, but still reaches a bag-of-words accuracy of $93.1\%$ accuracy for GPT-2, even for an update containing 13824 tokens.

This first insight already appears to have been overlooked in optimization-based attacks (Deng et al., 2021; Dimitrov et al., 2022) and does not require any malicious modifications of the embedding sent by the server. However, for a model update averaged over multiple users and thousands of tokens, this recovery alone is less of a threat to privacy. Naively, the ways to order recovered tokens scale as $n!$, and any given ordering of a large amount of tokens may not reveal sensitive information about any constituent user involved in the model update.

## D   MEASUREMENT NORMALIZATION

In order to retrieve positionally encoded features from the first linear layer in the feed-forward part of each Transformer block, we create "bins" which partition the possible outcomes of taking the scalar product of an embedding with a random Gaussian vector. To do this, we estimate the mean and variance of such a measurement, and create bins according to partitions of equal mass for a Gaussian with this mean and variance. A natural question arises: how well can we estimate this quantity? If the server does not have much data from the user distribution, will this present a problem for recovering user data? To this end, we demonstrate that the server can estimate these measurement quantities well using a surrogate corpus of data. In Figure 12, we see that the Gaussian fit using the Wikitext dataset is strikingly similar to the one the server would have found, even on a very different dataset (Shakespeare). We further note that an ambitious server could also directly use model updates

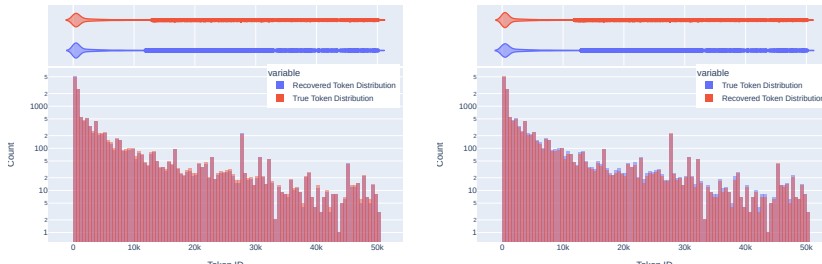

**Figure 11:** Bag-of-words Accuracy for from embedding gradients out of 13824 tokens from *wikitext*. **Left:** The small transformer model where tokens frequency can be estimated directly from the decoder bias. Token frequency estimation is 96.7% accurate, unique token recovery are 99.8% accurate. **Right:** The (small) GPT-2 variant where tokens are estimated from the norms of the embedding rows. Token frequency estimation is 93.1% accurate, unique tokens are 97.1% accurate due to the cutoff at $\frac{3\sigma}{2}$.

retrieved from users in a first round of federated learning, to directly glean the distribution of these linear layers, given that the lowest bin records the average of all activations of this layer. However, given the ubiquity of public text data, this step appears almost unnecessary - but possibly relevant in specialized text domains or when using Transformers for other data modalities such as tabular data (Somepalli et al., 2021).

However, for language data, these quantities can also simply be estimated using a small number of batches (100) filled with random token ids. We show in Figure 13 that this leads to essentially the same performance in average total accuracy and to only minor differences in peak accuracy.

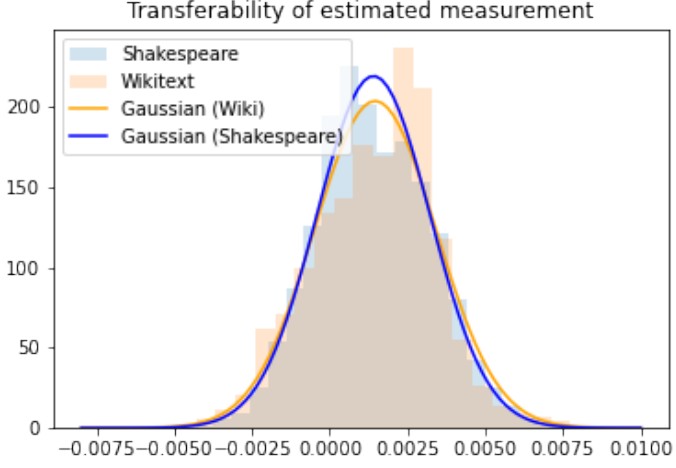

**Figure 12:** Comparison of Gaussian fit to the measurements of Shakespeare dataset, and a Gaussian fit to the measurements (against the same vector) for the Wikitext dataset.

### D.1    RANDOM TOKEN VULNERABILITY

Note that in Figure 8, we see that sequences generated randomly from the vocabulary are more vulnerable than other datasets. This is because randomly generated sequences on average result in fewer collisions as all possible tokens from the vocabulary are used with equal probability. Zipf's law for the distribution of words in English lends credence to this interpretation. More common words, and associated positions for that matter, will result in more collisions and degrade the attacker's ability to reconstruct sequences.

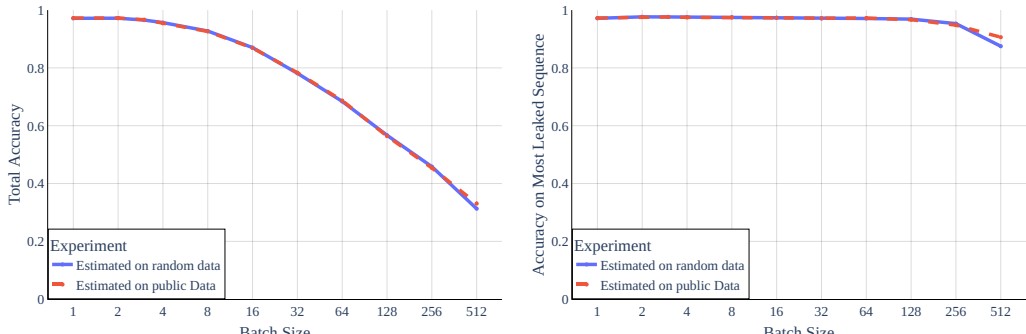

**Figure 13: Using random data to estimate the measurement distribution**. **Left:** Total accuracy (i.e. percentage of tokens recovered correctly in their correct position) for GPT-2 and varying batch sizes with sequence length 32. **Right:** Average token accuracy on the most-leaked sentence. Random tokens can be an effective substitute for public data to estimate the measurement mean and standard deviation.

# E  VARIANTS AND DETAILS

## E.1  MASKED LANGUAGE MODELLING

For problems with masked language modelling, the loss is sparse over the sequence length, as only masked tokens compute loss. This impacts the strategy proposed in Sec 3.2, as with disabled attention mechanisms in all but the first layer, no gradient information flows to unmasked entries in the sequence. However, this can be quickly solved by reactivating the last attention layer, so that it equally attends to all elements in the sequence with a minor weight increment which we set to 10. This way, gradient flow is re-enabled and all computations can proceed as designed. For BERT, we further disable the default `huggingface` initialiazation of the token embedding, which we reset to a random normal initialization.

Masked language modelling further inhibits the decoder bias strategy discussed in Sec 3.1, as only masked tokens lead to a non-positive decoder bias gradient. However, we can proceed for masked language models by recovering tokens from the embedding layer as discussed for models without decoder bias. The mask tokens extracted from the bias can later be used to fill masked positions in the input.

## E.2  GELU

A minor stumbling for the attacker occurs if the pre-defined model uses GELU (Hendrycks & Gimpel, 2016) activation instead of ReLU. This is because GELU does not threshold activations in the same was as ReLU, and transmits gradient signal when the activation $< 0$. However, a simple workaround for the attacker is to increase the size of the measurement vector and, by doing so, push activations away from 0, and thus more toward standard ReLU behavior. For the main-body experiments, we use ReLU activation for simplicity of presentation, but using the workaround described above, we find in Figure 14, Figure 15 that the performance of the attack against a GELU network is comparable to its level against a ReLU network.

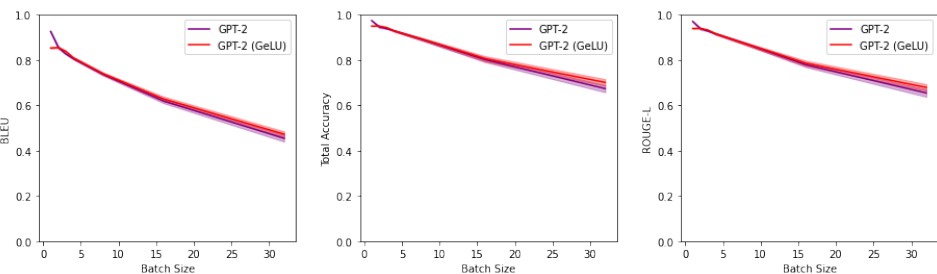

**Figure 14:** Comparison of GELU and ReLU activation with magnified measurement vector (sequence length 256).

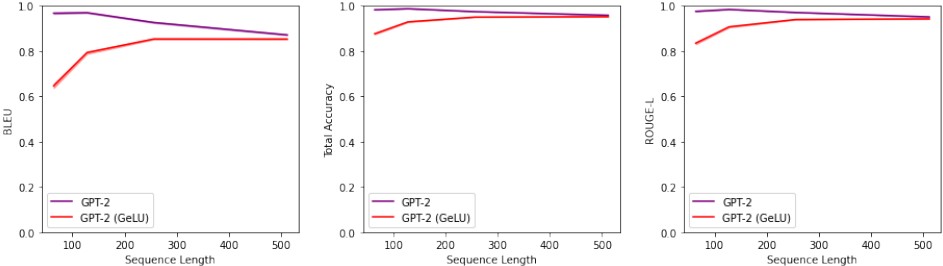

**Figure 15:** Comparison of GELU and ReLU activation with magnified measurement vector (batch size 1).

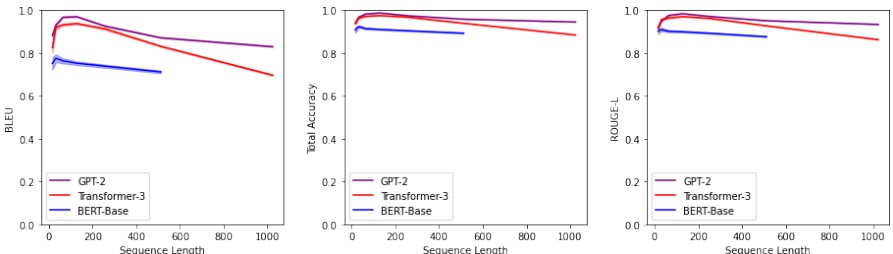

**Figure 16:** Baseline results for our method for different popular architectures and metrics for variable length sequence input (batch size 1). Note BERT's positional encoding caps out at sequence length 512, so we end our experiments for BERT there.

### E.3 DROPOUT

All experiments in the main body, including comparisons, have assumed that dropout has been turned off for all models under consideration. In standard applications, dropout can be modified from the server-side, even for compiled and otherwise fixed models (ONNX, 2022). In our threat model, dropout hence falls into the category of a server-side parameter that a malicious update could turn off. We also investigated the performance of the attack if the attacker cannot alter standard dropout parameters, finding that dropout decreased total accuracy of the attack by about 10%, e.g. from 93.36% for a sequence of length 512 on GPT-2 to 81.25% with dropout.

## F ADDITIONAL BACKGROUND MATERIAL

The attack TAG in Deng et al. (2021) approaches gradient inversion attacks against transformer models from the direct optimization-based angle. This was first suggested in Zhu et al. (2019), who provide some preliminary experiments on recovery of short sequences from BERT. Basically, the attack works to recover user data $(x^*, y^*)$ from the measurement of the gradient on this data $g$ by solving the optimization problem of

$$\min_{x,y} ||\mathcal{L}(n(x, \theta), y) - g||^2, \qquad (4)$$

where $x$ and $y$ are the inputs and labels to be recovered, $\mathcal{L}$ is the loss and $n(\cdot, \theta)$ is the language model with parameters $\theta$. This optimization approach does succeed for short sequences, but the optimization becomes quickly stuck in local minima and cannot recover the original input data. Zhu et al. (2019) originally propose the usage of an L-BFGS solver to optimize this problem, but this optimizer can often get stuck in local minima. Deng et al. (2021) instead utilize the "BERT" Adam optimizer with hyperparameters as in BERT training (Devlin et al., 2019). They further improve the objective by adding an additional term that especially penalizes gradients of the first layers, which we also implement when running their attack. A major problem for the attacks of Zhu et al. (2019) and Deng et al. (2021), however, is the large label space for next-word prediction. In comparison to binary classification tasks as mainly investigated in Deng et al. (2020), the large label space leads to significant uncertainty in the optimization of labels $y$, which leads their attack to reduce in performance as vocabulary size increases.

We further note that Boenisch et al. (2021) also propose malicious model modifications (as opposed to our more realistic malicious parameter modifications) to breach privacy in FL for text, however the

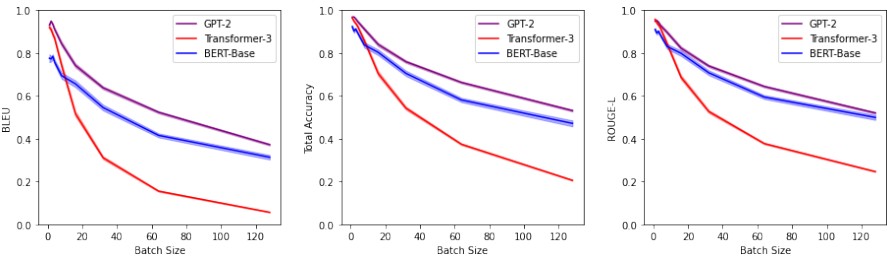

**Figure 17:** Results for our method for different popular architectures and metrics for variable length batch size (sequence length 32).

proposed model in their work is a toy two-layer fully connected model that is not a Transformer-based model. In fact, the strategy employed in their attack cannot be deployed against modern language models that do not construct a linear layer of the length of the sequence, aside from handling of facets of modern language models like positional encodings, or attention mechanisms.

# G  MITIGATION STRATEGIES

As stated in the main body, a central message of our paper is that aggregation alone may not be sufficient to defend against malicious parameter attacks in FL. Further defenses may be needed. Such defenses could include parameter inspection wherein a defender could look for identifiable signatures in the shared parameter vector. For example, the given attack duplicates a measurement vector, leading to low rank linear layers. To the best of our knowledge, there do not exist systematic parameter inspection based defenses to malicious models in FL. The attacker could, however, easily adapt to some obvious parameter inspection defenses. For example, simply adding a small amount of noise to the measurement vector, and other conspicuous parts of the attack, does not affect the success of the attack significantly, but does immediately make a strategy like rank inspection null and void. We confirm this adaptation by adding a small amount of Gaussian noise to each row of the measurement vector in the linear layers (for the Transformer-3 model, 8 users, 32 sequence length) and find that numerically, these layers do indeed become full rank, and the success of the attack remains largely unchanged (accuracy 90.62%). Because such simple adaptations are easily made, we do not recommend this line of defense.

Instead, we advocate for strong differential privacy on the user side. Such measures could include gradient clipping/noising (c.f. .

The immediate impact of differential privacy mechanisms on the attack described in this work is that the measured weight and bias become noisy, so that the estimation in Equation (3) becomes subsequently noisy. However, an attacker can counteract this simply by noticing that the number of jumps in the cumulative sum is sparse, i.e only number-of-tokens many jumps exist, independent from the number of bins. As such, as long as the number of tokens is smaller than the number of bins, the attacker can denoise this measurement, for example using tools from sparse signal recovery such orthogonal matching pursuit (Mallat & Zhang, 1993). We use the implementation via K-SVD from Rubinstein et al. (2008) to do so. This way, the attacker reduces the immediate effect of gradient noise, utilizing the redundancy of the measurements of the cumulative sum of embeddings in every bin. Overall, with the above strategy, we find the attack can still succeed in the presence of gradient clipping and noising, although the success does predictably degrade at higher noise levels. (see Figure 18).

Furthermore, the imprint strategy, as found in Fowl et al. (2021), has been shown to be robust to certain amounts of gradient noise, and an attacker could further attempt to implement strategies found in that attack to resist this defense.

**Mitigations through local loss audits**  Moreover, federated learning often takes place in the background, for example, at night when the device is charging (see Hard et al. (2019)) and can be a regular occurrence on participating devices. Even if user applications were modified to make loss values on local data observable to the user, a user could not know whether a model was malicious, or whether a new model was simply in an early round of its training run. As such, auditing and exposition of local model loss values to the user is not a strong defense against the proposed attack.

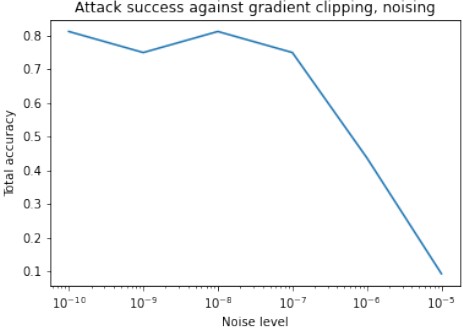

**Figure 18:** Attack success rate against different levels of added Laplacian noise. In each experiment, the gradient was also clipped to a value of 1. We see that the attack can remain successful even in the presence of added noise. In this experiment, a single sequence of length 32 was considered.

# H FURTHER RESULTS

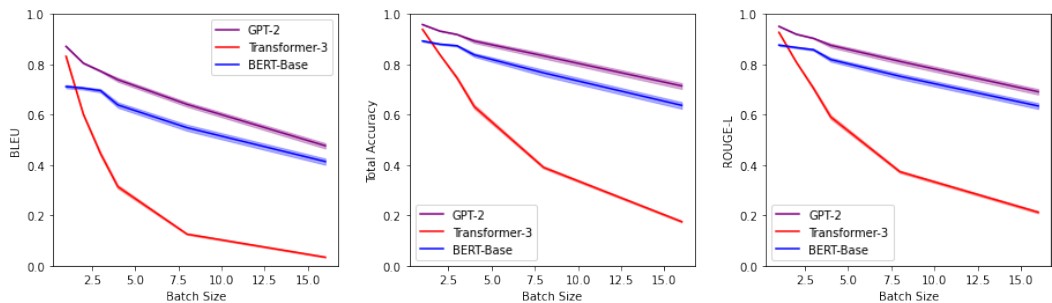

**Figure 19:** Baseline results for our method for different popular architectures and metrics for variable batch size (sequence length 512).

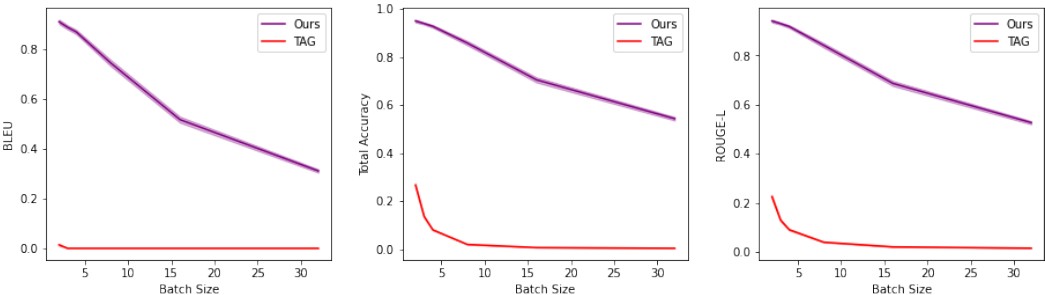

**Figure 20:** Comparison to TAG attack for variable batch size (sequence length 32).

In Figure 3, we illustrate the embedding recovery process that we adapt from Fowl et al. (2021). We opt to initialize the rows of the first linear layer in each Transformer block to a randomly sampled Gaussian vector (all rows being the same vector). Then, following the calculation of Fowl et al. (2021), when an incoming embedding vector, for example, corresponding to the word "pepper", is propagated through this layer, the ascending biases threshold values of the inner product between the Gaussian vector, $m$, and the embedding vector, $x$. Because of the ReLU activation, this means that individual embedding vectors can be directly encoded into gradient entries for rows of this linear layer.

## H.1 SENSITIVITY TO THRESHOLD IN TOKEN ESTIMATION

We include an evaluation of the threshold parameter for the embedding-norm token estimation in Figure 21. This value is set to 1.5 standard deviations in all other experiments.

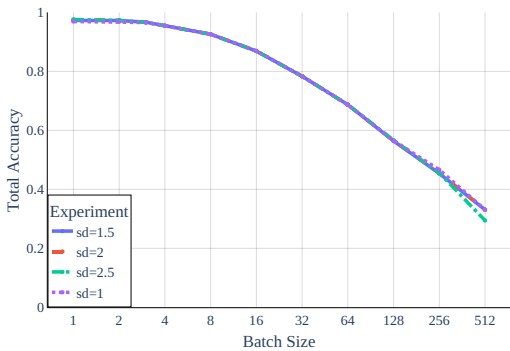

**Figure 21: Sensitivity to token estimation threshold.** Total accuracy (i.e. percentage of tokens recovered correctly in their correct position) for GPT-2 and varying batch sizes with sequence length 32. We evaluate the threshold parameter for the token estimation. The default value in the remainder of all experiments is 1.5, although we find the estimation to be robust to a range of similar cutoffs.

## H.2 TOTAL AMOUNT OF TOKEN LEAKED

Finally, in Figure 22 we include the total amount of tokens leaked correctly through the attack for a variety of sequence lengths, batch sizes and models. We see that the total amount of tokens leaked with GPT-2 has not yet reached a peak, and larger sequences could leak more tokens. For the smaller 3-layer transformer we see that the number of leaked tokens peaks around a batch size of 128 for a sequence length of 32, and 8 for a sequence length of 512, i.e. around 4096 tokens. This is related to the number of bins in this architecture, which in turn is given by the width of all linear layers, leading to a bin size of 4608 for the 3-layer transformer. For the small GPT-2 the number of bins is 36864.

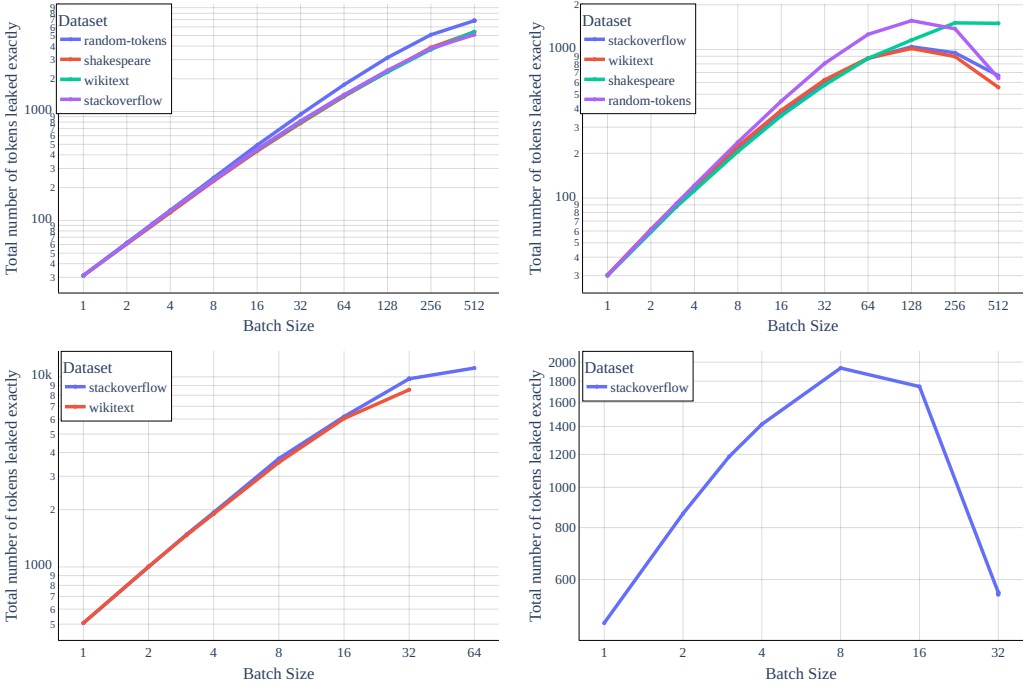

**Figure 22: Total number of tokens leaked in their exact location for various settings (instead of percentages as shown elsewhere).** Top row: Total number of token for GPT-2 (left) and the 3-layer transformer (right) for a sequence length of 32 and various batch sizes. Bottom row: total number of token for GPT-2 (left) and the 3-layer transformer (right) for a sequence length of 512 and various batch sizes.

