# OpenReview forum: "Decepticons: Corrupted Transformers Breach Privacy in Federated Learning for Language Models"
_ICLR.cc/2023/Conference — ICLR 2023 poster_

### Official Review · Reviewer_Fg6a · 2022-10-28

**Confidence:** 4
**Correctness:** 4
**Technical Novelty And Significance:** 3
**Empirical Novelty And Significance:** 3
**Recommendation:** 8

**Clarity, Quality, Novelty And Reproducibility:**

Clarity: The paper is clearly written and easy to follow.

Quality: The overall quality of the paper is quite high. The provided figures illustrate the proposed concepts and make it easier to understand the approach. However, the font size in some figures (e.g., Fig. 2) is a bit small, and increasing it would improve readability.

Novelty: Whereas some parts of the attacks seem to be already known, the attack combines different approaches and adds its own developments to them. Also, a novel threat setting is introduced, which opens an interesting avenue for future research.

Reproducibility: Hyperparameters, model and dataset details and source code are provided. I did not run the code but I expect at least the reproduction of the attack is possible.

Some concerns regarding the limitations: The attacker first updates the model weights of a victim to deactivate the attention layers and most feed-forward layers to make gradient inversion and token reconstruction feasible. The victim then performs an update, which is sent to the server and inverted. So let us assume the model is used for something like keystroke prediction or translation. Whereas the model should behave as expected on benign weights, after replacing the model weights with the malicious server state, the model should in this particular state basically be useless in terms of prediction performance. I assume that any user then will then note this significant drop in model performance, either by a strong increase in the training loss or because the model produces meaningless outputs. So the attack is not as secretly performed as other attacks that are only based on the gradient updates and do not alter the victim's model. If this is true, it should at least be discussed in the paper as a limitation and maybe also a possible attack detection. However, I do not think this fact affects the overall contribution of the paper in a significantly bad way.

Small remarks:
- The term "fedAVG" should be introduced with a small sentence for readers not familiar with the term.
- Fig. 7: The "Loading MathJax" in the images should be removed.
- The ethics statement mainly discusses possible mitigations. However, a discussion on potentially negative impacts on the security and privacy of models and user data is missing.

**Details Of Ethics Concerns:**

The paper proposes a new attack against federated learning that might have a negative impact if applied in real-world applications. Whereas the paper does not propose a new defense, some approaches from the literature are discussed in the "Ethics Statement" section.

**Strength And Weaknesses:**

Strengths:
+ clearly written paper, which is easy to follow. Especially the given intuitions and visualizations of the attack are helpful for understanding.
+ interesting new attack setting (server sends poisoned model weights to facilitate the attack)
+ approach combines various ideas of previous research but also offers its own improvements
+ empirical results state strong attack success

Weaknesses:
- The interpretation of the empirical results could use a little more detail. For example, the paper shows that random data is more vulnerable than natural language inputs. This is a very interesting finding, and discussing possible reasons would improve the paper.
- I miss some discussion of the limitations of the approach, particularly the fact that the model after the malicious update won't produce any meaningful predictions on the user side (see next section for more details)

**Summary Of The Paper:**

The paper introduces a novel gradient inversion attack against NLP transformer models in a federated learning setting. The attack is constructed around a new threat model that allows the attacker (server) to deploy the model on user devices with poisoned model parameters, which are downloaded from the server and replaced by benign parameters with the next update. This poisoned model update disables the attention layers and (most) FF layers to facilitate the separation and recovery of input tokens. The proposed attack is then evaluated on different transformer-based models of varying model sizes.

**Summary Of The Review:**

I like the proposed attack based on the idea of facilitating gradient inversion attacks by applying a malicious update to the victim model. It offers a new perspective to the research in this area and is well presented throughout the paper. I only have small concerns with the rather short interpretation of the evaluation results. Overall, I think the community will benefit if the paper will get accepted.

---

> ### Author Response · Authors · 2022-11-12
> **Response**
>
> Thank you for your time and thoughtful review. Below we respond to the points you raise.
>
> &nbsp;
>
> > some figures (e.g., Fig. 2) is a bit small, and increasing it would improve readability.
>
> Thank you for pointing this out. We have increased the font of the labels in Figure 2, 4. Please let us know if anything else could use magnification.
>
> &nbsp;
>
> > …  For example, the paper shows that random data is more vulnerable than natural language inputs.
>
> Thank you for raising this question. We have since added clarification in Appendix Section D.1. To explain briefly, we hypothesize that randomly generated sequences are more vulnerable to our attack because these sequences on average result in fewer collisions as all possible tokens from the vocabulary are used with equal probability.
>
> In “natural” language, words appear with non-uniform frequency (c.f. Zipf's law). For example, the word “the” will tend to appear more than the word “platypus”. In a batch of sequences, this can lead to collisions for the attacker as the token embeddings for the repeated words are identical. This is related to where in the density the measurement falls in Figure 3.
>
>
> &nbsp;
>
> > particularly the fact that the model after the malicious update won't produce any meaningful predictions on the user side
>
> Please see General Comment 1 for our response. We note in summary that the model is not required to be functional, and a high loss value could simply reflect a model in the early stages of training. Moreover, we generally consider the case where a single global model is trained by all devices, only later deployed to all users (as is the case in Hard et al. 2019). Such a model (with multiple configurations and variations) can be in training in successive rounds on a given pool of users, and can be in any stage of training when received by a device.
>
> &nbsp;
>
> > The term "fedAVG" should be introduced with a small sentence for readers not familiar with the term.
>
> > Fig. 7: The "Loading MathJax" in the images should be removed.
>
> Thank you for bringing this up. We’ve now clarified this acronym in our writing and fixed this figure.
>
> &nbsp;
>
> > The ethics statement mainly discusses possible mitigations. However, a discussion on potentially negative impacts on the security and privacy of models and user data is missing.
>
> We’ve amended our ethics statements to include a description of potential negative impacts. In summary, we believe that an attack like the one we describe poses a serious risk to sensitive user text data. We hope that our work raises awareness of such attacks and leads to more strict protocols and defenses employed by the community and companies.

---

> > ### Comment · Reviewer_Fg6a · 2022-11-14
> > **Thank you for the feedback**
> >
> > I thank the authors for clarifying my concerns, providing detailed answers to my questions, and improving the readability of the paper. Overall, I still think this is a good paper that deserves to get accepted. Therefore, I will stick to my initial rating.

---

### Official Review · Reviewer_eNPF · 2022-10-28

**Confidence:** 5
**Correctness:** 2
**Technical Novelty And Significance:** 1
**Empirical Novelty And Significance:** 1
**Recommendation:** 1

**Clarity, Quality, Novelty And Reproducibility:**

The paper is well-written, and I enjoy reading the paper. The implementation is available, indicating that the result can be reproduced with some effort. However, the paper does not advance my knowledge in data reconstruction attacks v.s. privacy-preserving and model utility in real-world applications. It appears that the paper will bring more confusion to the community instead of answering already-known but open questions.

**Details Of Ethics Concerns:**

The paper discusses potential mitigations for the proposed privacy inference attacks. There is no concern from my view.

**Strength And Weaknesses:**

Strength

+ The research topic is critical.
+ The proposed attack is somewhat new for text data.

Despite an important topic, the paper has several crucial problems, as follows:

Weaknesses

- The federated learning setting in this paper is unrealistic. There is no evidence to support that clients train BERT or Transformers on their local devices for keystroke prediction tasks in real-world applications (deployment of such a setting). Training BERT or Transformers requires computational power on the training devices. It will incur communication costs for the local devices as well. Assuming local devices could be personal computers or mobile devices, how practical is the setting in the real world? Are there any deployed systems for such applications? If this is a cross-silo setting, what is the benefit of using FL compared with centralized training independently among clients?

- The attacks appear to be easily detectable since many model weights are set to either 1 or 0. For instance, the $W_K$ matrix is set to identity 1, $b_K = 0$, $W_ Q = 0$, $W_V=1$, and $b_v = 0$. In the real world, when such conditions are blended into the actual parameter distribution making the attack stealthy? If clients check one of these conditions, they can easily ignore the training round, thus avoiding the attack. That poses a fundamental question: How do we theoretically and empirically understand the stealthiness of the proposed attack? In addition, the attacker updates the model weights to deactivate the attention and feed-forward layers. That introduces a significant gap between model utility before, during, and after the attack. This gap can be visible to the clients to detect the attack. How could the proposed attack avoid or minimize this gap? What is the fundamental advantage to make the attack severe?

- The paper does not provide any empirical or theoretical analysis of the proposed attack against privacy-preserving mechanisms, such as DP and local DP. It is unrealistic to assume that the clients naively follow the protocol set up by the server. Therefore, a data reconstruction attack needs to be evaluated against privacy-preserving mechanisms. The critical point is not the attack but the trade-off between the attack, privacy protection, and model utility. That will inform the practicability of the attack in real-world applications.

**Summary Of The Paper:**

This paper presents a new data reconstruction attack for text data in federated learning. In the proposed attack, a dishonest server drafts corrupted model parameters and sends them to the clients. The server can reconstruct sequences of text by observing the gradients returned by the clients using the corrupted model parameters. The proposed attack is tailored to small Transformers. Experimental results were conducted on several text datasets, including wikitext, Shakespeare, and StackOverflow.

**Summary Of The Review:**

The paper studies a fundamental data reconstruction problem for text data in federated learning. However, there is room for improvement.

---

> ### Author Response · Authors · 2022-11-12
> **Response (2/2)**
>
> >The attacks appear to be easily detectable
> We discuss this in the “Mitigations” section of our manuscript. To summarize:
>
> Parameter inspection (on the server side) is not implemented in any major industrial FL framework (See Paulik et al. 2021, Dimitriadis et al. 2022, Li et al., 2019, Bonawitz et al. 2019). To further highlight this point, these are the published frameworks for federated learning for Apple (Paulik), Microsoft (Dimitriadis), Nvidia (Li), and Google (Bonawitz).
>
> Moreover, even if users could get access, and “inspect” the model updates sent by the server, there remain several problems:
>   * Knowledge of our attack is necessary to know what to look for - which is why we encourage publication of this attack in this work.
>   * Ostensibly, the inspection system would be automated. Would such a system just check for the conditions you list? The user in this case is always playing “catch-up” and the attacker could very simply modify the attack to evade detection, given that the attacker here is the server with full knowledge of the detection mechanism. We include experimentation on this in Section G of our appendix, and find that adding a small amount of noise to the malicious parameters (in order to make them full rank) does not degrade the attack.
>   * The model updates are ephemeral - an attacker could simply send benign updates, then an instantaneous privacy attack, then more benign updates.
>
> &nbsp;
>
> > … significant gap between model utility before, during, and after the attack. This gap can be visible to the clients to detect the attack.
>
> Please see General Comments 1. For a user, the loss of a model very early in training would be indistinguishable from the loss during an attack.
>
> &nbsp;
>
> >The paper does not provide any empirical or theoretical analysis of the proposed attack against privacy-preserving mechanisms
>
> We do in fact provide empirical analysis of gradient clipping and noising in Section G of our Appendix (see Figure 18). We have modified the main body of our work to now more clearly direct readers to the relevant Appendix section.
>
> We do agree though that stronger defenses are needed, and revealing attacks like ours puts pressure on federated learning practitioners to employ stronger techniques - like DP based defenses.
>
> &nbsp;
>
> > It is unrealistic to assume that the clients naively follow the protocol set up by the server.
>
> None of the currently implemented systems allow users to modify the inner workings of the FL protocol on their own.. Users generally download and receive verified packages from OS maintainers that execute FL payloads on their devices. Even deviating from this protocol would require a user to “root” their own device and intercept and replace messages going to the FL system - at which point the user would have gotten much further by simply opting out of the system.
>
> Besides this setup, also described in many works such as Ramaswamy et al. 2019  and Hartmann et al. 2021, what system design of FL are you thinking of? There is no evidence that users in *actual* federated systems are anything but willing and “naive” participants in the federated protocol.
>
> Further,  on the server side, we know that most federated systems currently in production do not use any DP mechanisms (see Hard et al. 2019, [6], [7]). In these cases, secure aggregation might be the only “defense” against privacy attacks [7].
>
> &nbsp;
>
> Do you have any further concerns? We’re happy that you enjoyed reading this paper and consider this research topic critical for privacy in FL. We hope that the above clarifications convince you to improve your score from your initial “strong reject” recommendation.
>
> &nbsp;
>
> [1] Yang, Timothy, et al. "Applied federated learning: Improving google keyboard query suggestions." arXiv preprint arXiv:1812.02903 (2018).
>
> [2] Rieke, Nicola, et al. "The future of digital health with federated learning." NPJ digital medicine 3.1 (2020): 1-7.
>
> [3] Geiping, Jonas, et al. "Inverting gradients-how easy is it to break privacy in federated learning?." Advances in Neural Information Processing Systems 33 (2020): 16937-16947.
>
> [4] Yin, Hongxu, et al. "See through gradients: Image batch recovery via gradinversion." Proceedings of the IEEE/CVF Conference on Computer Vision and Pattern Recognition. 2021.
>
> [5] Zhu, Ligeng, Zhijian Liu, and Song Han. "Deep leakage from gradients." Advances in neural information processing systems 32 (2019).
>
> [6] Chen, Mingqing, et al. "Federated learning of out-of-vocabulary words." arXiv preprint arXiv:1903.10635 (2019).
>
> [7] https://support.google.com/messages/answer/9327902

---

> > ### Comment · Reviewer_eNPF · 2022-11-17
> > **Unconvincing threat models - attacks - and defenses. My concerns have not been fully addressed.**
> >
> > Thanks a lot for the response from the authors. The response does not address my concerns regarding the realistic picture of threat models - attacks - and defenses.
> >
> > In the first response, there is no evidence that BERT or Transformers can be trained on mobile devices in recently developed federated learning frameworks and system deployments. From my knowledge, training BERT or Transformers  on mobile devices is infeasible at the moment. Second, why do we need "cross-silo" for keystroke prediction tasks? The benefit of FL is unclear here.
> >
> > Second, a small amount of noise injected to the malicious parameters does not present state-of-the-art defenses against these types of attacks. The authors also agree that stronger defenses are needed to reflect a better understading of threat models - attacks - and defenses.
> >
> > Third, from privacy and security point of views, I agree with the authors that the clients can root their devices to get the models. This can be done by a trusted inspection system offered to the clients. The point is: It is hard to say that there is no way for the clients can inspect the models, which are stored and operated in their local devices. If so, we do not need to study compromised clients, who can intercept the federated training, at all in FL. Why don't clients use similar approaches with the compromised clients to inspect the model? The attack algorithms need to ensure that given the presence of such inspection system, the attacks can still be successful. The paper does not address this point.
> >
> > Fourth, "most federated systems currently in production do not use any DP mechanisms" does not mean that they are not going to use privacy-preserving mechanisms. For instance, I found one paper using Local DP in their FL system design [1]. Many other papers use privacy preserving-FL as well. We cannot urgue that FL frameworks will not apply privacy preserving mechanisms in their systems. The point is: Without using privacy-preserving mechanisms, there is no privacy protection for the clients' local data. Everyone knows this, and the clients know this as well. Therefore, there is no point of providing a FL system without privacy protection against potential privacy risks? Is this a realistic application of FL in the real world?
> >
> > Overall, I enjoy reading the paper and thanks a lot for the authors's response. I do appreciate all the works from the authors. However, a threat model/attack needs to take into account realistic defenses to really understand the attack's impacts in a real-world setting. A throrough study regarding the correlation among threat models - attacks - and defenses is needed. I would love to see how the attacks hold up against defense mechanisms rather than one-sided attack results. That is where the attack really advances the state-of-the-art.
> >
> > Therefore, I keep my score at this moment.
> >
> > Sinrecely,
> >
> > [1] https://www.researchgate.net/publication/356375862_FLSys_Toward_an_Open_Ecosystem_for_Federated_Learning_Mobile_Apps

---

> > > ### Comment · Reviewer_eNPF · 2022-11-17
> > > **Additional Concern**
> > >
> > > I am thinking a lot here. The proposed attack is only effective at initial training rounds when the model loss is still high. If training BERT and Transformers on mobile devices from scratch, how much is the cost for the clients to pay, i.e., how many iterations, communication and computation costs, data plan, etc., to achieve usable models? It could take years for this. Would the clients be willing to do that? To avoid this burden, the server or service providers need to use a pre-trained model to boost the utility before fine-tuning it on the clients' local data. In that case, the attack becomes ineffective since the model loss would be low initially. That exposes the attacks (having significant high model loss) to be detected by the clients.
> > >
> > > So, what are the practical settings for the attack to be launched in real-world applications?

---

> > > > ### Author Response · Authors · 2022-11-17
> > > > **Response to Additional Concern**
> > > >
> > > > >I am thinking a lot here. The proposed attack is only effective at initial training rounds when the model loss is still high
> > > >
> > > > The attack is effective at any point at which the malicious parameters are deployed to the user devices. Are you assuming again a loss inspection defense? If so, please do read through our previous response to such a defense. Given that federated models are not immediately deployed to user devices, why does the model having a large loss matter at all?
> > > >
> > > > &nbsp;
> > > >
> > > > >how much is the cost for the clients to pay, i.e., how many iterations, communication and computation costs, data plan, etc., to achieve usable models
> > > >
> > > > We’re confused by your point here. Are you wondering whether FL could ever practically be deployed because of potential electricity costs to clients? FL is already deployed, and generally incurs negligible costs on users. In terms of time necessary for the attack, this is also negligible, and could happen in a matter of seconds, after which a benign set of parameters could be sent to the users to resume standard training.
> > > >
> > > > &nbsp;
> > > >
> > > > >To avoid this burden, the server or service providers need to use a pre-trained model to boost the utility before fine-tuning it on the clients' local data.
> > > >
> > > > This is not the case. Random initializations are often used - please see Hard et al. 2019. Even here, given the amount of users who participate, training actually finishes in a few days.

---

> > > > > ### Comment · Reviewer_eNPF · 2022-11-17
> > > > > **Local DP Result?**
> > > > >
> > > > > Where could I find the attack results against Local DP? I do not see it in section G. Note that gradient clipping and noise injection does not preserve local DP. What is the level of privacy protection offered by your Local DP?
> > > > >
> > > > > P/s: Thanks a lot for your response. I am looking for the lanscape of the proposed attacks v.s. defenses (such as Local DP). It will be convincing if such results are included in the paper.
> > > > >
> > > > > Sincerely,

---

> > > > > > ### Author Response · Authors · 2022-11-18
> > > > > > **Response**
> > > > > >
> > > > > > Thank you for your continued engagement! There may be confusion (caused by our wording) here. We are taking cues from McMahan et al. 2018 for their DP-FedSGD setup, and we apologize for confusion over “local” vs. “user-level” DP - we have loosely used these terms and we did not mean to conflate them. We are primarily interested in user-level DP mechanisms.
> > > > > >
> > > > > > &nbsp;
> > > > > >
> > > > > > >What is the level of privacy protection?
> > > > > >
> > > > > > We provide results for the sensitivity of the attack to a Laplacian mechanism in Fig.18 in the Appendix. We think this would be most useful as a study of the effect of the attack against such a mechanism, simulating the setting of McMahan et al. 2018. The actual privacy guarantee enforced by this setup through composition would depend further on other hyperparameters, such as sampling rates, amount of users and possibility of resampling, which would be application-specific.
> > > > > >
> > > > > > &nbsp;
> > > > > >
> > > > > > >I am looking for the lanscape of the proposed attacks v.s. defenses
> > > > > >
> > > > > > We appreciate your concern here, and we are actually in agreement that stronger DP-based defenses are needed against these attacks. We would like to emphasize one more time that the purpose of our work is to demonstrate (the first) empirical attack against a federated language system with secure aggregation. This is vitally important because, although secure aggregation offers no formal privacy guarantee, it is currently believed to offer sufficient empirical privacy to users - as evidenced by the numerous production systems for which this is the only privacy defense implemented (see previous responses).
> > > > > >
> > > > > > Previous attacks against FL systems [Zhu et al. 2019, Deng et al. 2021, Geiping et al. 2020, Yin et al. 2021, Gupta et al. 2022, etc.] fail predictably when aggregation is employed, and, as such, do not include any results regarding the sensitivity to noise.
> > > > > > We do want to reiterate the big picture is that these previous attacks present a “false sense of security” of aggregation as a defense. It is this notion that we want to correct with the attack discussed in our work.
> > > > > > We agree with the reviewer that stronger defenses are required to make these systems secure, and we are confident our work provides an example of why that is.
> > > > > >
> > > > > > &nbsp;
> > > > > >
> > > > > > We hope you could agree that this example is worthwhile for the community, given the litany of other works that induce this “false sense of security”.

---

> > > > > > > ### Comment · Reviewer_eNPF · 2022-11-18
> > > > > > > **Figure 18 is not appropriately considered a suitable defense.**
> > > > > > >
> > > > > > > I looked into Figure 18 before. This figure triggers a lot of confusion while providing no meaningful defenses. I did not buy it, and that is why I asked questions.
> > > > > > >
> > > > > > > First, User-DP does not defend against reconstruction attacks.
> > > > > > >
> > > > > > > Second, the threat model considers a dishonest server. Including the results of User-DP is meaningless since the server can ignore the noise injection. In addition, the noise scale is too small, i.e., 10^-5, compared with the clipping bound, i.e., 1. Therefore, no defense at all is provided in Figure 18. What is the value of the privacy budget epsilon in User-DP offered with such a level of noise scale and clipping bound? How much the model utility is affected? The attacks still failed to reconstruct the data under such a loose condition.
> > > > > > >
> > > > > > > There is a simple question: If we can totally prevent the attacks with a trivial defense (even with no realistic privacy protection), what is the point of launching the attacks?
> > > > > > >
> > > > > > > Third, there are works in literature to preserve Local DP in training NLP models. A convincing landscape of an attack v.s. defenses will truly advance our knowledge by carefully considering the state-of-the-art. That brings true value to help us in developing practical solutions in real-world applications. Otherwise, I am afraid that the reported results will mislead us.
> > > > > > >
> > > > > > > One setting showing that the attack could bypass a convincing privacy-preserving defense would be helpful. I will lift my score if such evidence shows that the attack is severe.
> > > > > > >
> > > > > > > P/s: I thanks the authors for your responses. I do appreciate it.
> > > > > > >
> > > > > > > Sincerely,

---

> > > > > > > > ### Author Response · Authors · 2022-11-18
> > > > > > > > **Response**
> > > > > > > >
> > > > > > > > Thank you for your thoughts and ideas on this.
> > > > > > > >
> > > > > > > > &nbsp;
> > > > > > > >
> > > > > > > > > Including the results of User-DP is meaningless since the server can ignore the noise injection.
> > > > > > > >
> > > > > > > > In the case where it is strictly on the server side to choose whether/how much noise to add, then we agree that such a defense becomes meaningless.
> > > > > > > >
> > > > > > > > Our analysis is  in the context of  a scenario where noise is strictly added on the user device before the update is communicated to the server. We agree that this will only provide a level of privacy that can be computed a-posteriori, because the amount of noise on the user side has to be fixed beforehand and would be independent from other contributions.
> > > > > > > >
> > > > > > > > &nbsp;
> > > > > > > >
> > > > > > > > >  In addition, the noise scale is too small, i.e., 10^-5, compared with the clipping bound, i.e., 1.
> > > > > > > >
> > > > > > > > Can you provide additional references for this conclusion? Again, without hyperparameters, it is quite difficult to consider formal bounds. In McMahan et al. 2019 (with a different architecture than ours), for example, a noise level of $\sim 10^{-3}$ is considered in experiments, but training also occurs over thousands of rounds, incurring additional privacy loss. Whereas our attack could take place in a *single* round.
> > > > > > > >
> > > > > > > > &nbsp;
> > > > > > > >
> > > > > > > > > Third, there are works in literature to preserve Local DP in training NLP models.
> > > > > > > >
> > > > > > > > Under secure aggregation, the server does not have access to *individual* responses, only aggregated ones. Generally speaking though, while promising results have been shown in limited cases for maintaining accuracy while training with strong DP mechanisms, these methods have generally not been adopted in real-world FL for fears of quality degradation.
> > > > > > > >
> > > > > > > > &nbsp;
> > > > > > > >
> > > > > > > > > A convincing landscape of an attack v.s. defenses will truly advance our knowledge by carefully considering the state-of-the-art.
> > > > > > > >
> > > > > > > > We argue that a convincing landscape instead considers what is *actually* used in current FL setups. The message and value of our paper is simple: current attacks against privacy in FL on text fail when exposed to even modest aggregation. This leads many to believe that aggregation is a sufficient defense - as evidenced by Hard et al. 2019, [6], [7], and many other applications applying only secure aggregation as a defense. We reveal an attack which demonstrates that this empirical assurance of privacy is insufficient. *We are quite interested to know what part of this argument you view as not worthy of publication?* Especially considering that we go above and beyond the list of previously published, and highly cited attacks in terms of defenses tested against.
> > > > > > > >
> > > > > > > >
> > > > > > > > As a general point, we’re not quite sure what the reviewer aims to find? We know that DP mechanisms provably degrade recovery attacks by nature of the attack. We have no doubt that under sufficient noise levels, our attack, as well as every other attack in the field, will degrade, which is why we do agree that further defenses, possibly DP-based, are needed.
> > > > > > > >
> > > > > > > > Yet, even if the take-away would be that small levels of DP protects against an attack like this, it is still very valuable that our work empirically points to the necessity of this defense via DP. Previous attacks in FL have *not* shown this necessity.

---

> > > ### Author Response · Authors · 2022-11-17
> > > **Response (2/2)**
> > >
> > > > Fourth, "most federated systems currently in production do not use any DP mechanisms" does not mean that they are not going to use privacy-preserving mechanisms. For instance, I found one paper using Local DP in their FL system design [1].
> > >
> > > We cite several actually deployed FL systems which do not use any form of DP. Is your point that it is possible that a system could deploy DP mechanisms? If so, we of course agree - which is why we said *most* do not use it currently, which we hope you agree with after seeing our references of highly cited and actually deployed FL systems.
> > > Also, we would like to stress that we actually *do include* empirical experimentation against local DP mechanisms (above and beyond what the previous attacks we cite do).
> > >
> > > &nbsp;
> > >
> > > >  We cannot urgue that FL frameworks will not apply privacy preserving mechanisms in their systems
> > >
> > > We can in fact argue this as many actually deployed systems use no DP mechanisms. Please see  Hard et al. 2019, [6], [7].
> > >
> > > &nbsp;
> > >
> > > >Without using privacy-preserving mechanisms, there is no privacy protection for the clients' local data. Everyone knows this, and the clients know this as well. Therefore, there is no point of providing a FL system without privacy protection against potential privacy risks?
> > >
> > >
> > > To the best of our knowledge, we are actually the first work to empirically demonstrate that secure aggregation is an insufficient defense to FL for language tasks. More importantly, this claim doesn’t hold water when several uses of FL rely solely on aggregation to promise users privacy (Hard et al. 2019, [6], [7]).  To this point, we are surprised that “Everyone knows this”, and we would gladly join you in letting system providers like Google messages know that their promise “To ensure your data is kept private and secure, the federated technologies use Secure Aggregation” [2] is useless given this attack!
> > >
> > >
> > > &nbsp;
> > >
> > > > However, a threat model/attack needs to take into account realistic defenses to really understand the attack's impacts in a real-world setting. A throrough study regarding the correlation among threat models - attacks - and defenses is needed.
> > >
> > >
> > > We explicitly discuss the merits of mitigation strategies in our paper, see page 10.
> > > In this work, we show practical attacks against realistic (here, meaning real systems) defenses (secure aggregation) that are deployed in real-world contexts (Hard et al. 2019, [6], [7]).
> > > To be more explicit, if an attack like ours would have been deployed in any of the above known FL use cases - *exactly* how they are described, user data would have been compromised - Given this, we are curious about how more “real-world” the attack could be?
> > >
> > > &nbsp;
> > >
> > > Please let us know if we can clarify any further confusions!

---

> > > ### Author Response · Authors · 2022-11-17
> > > **Response (1/2)**
> > >
> > >
> > >
> > > Thank you for the reply.
> > >
> > > &nbsp;
> > >
> > > > In the first response, there is no evidence that BERT or Transformers can be trained on mobile devices in recently developed federated learning frameworks and system deployments
> > >
> > > Were you able to read through the field guide for federated learning that we reference in both our manuscript and our response? In this field guide - produced by Google, *the industrial pioneer* of federated learning, the architecture which they use for all of their federated language experiments is the *same exact* architecture as the 3-layer Transformer on which we experiment. Why is this “no evidence”? The above model is only 13MB, which is easily trained on modern smartphones, especially when plugged in. This is well within the 2GB required space as referenced in Hard et al. 2019.  Are the other highly cited papers we reference, which experiment with even larger models, not evidence enough of concern and community interest about such attacks? We are honestly confused why you think of all of these are “no evidence”.
> > >
> > > Moreover, what is the broader implication of your concern here? Are you adamant that small transformers like the one we discuss in this work are categorically impossible to use in decentralized applications and will never be used on mobile devices?
> > >
> > >
> > > Furthermore, while we do consider mobile devices an interesting and motivating example where these types of attacks are possible, the proposed attack and our work  is not specific to this single scenario of mobile device federated learning. The attack can be mounted against any federated learning system using transformer-based architectures and any objective, not just next-word-prediction, as we discuss in our previous response.
> > >
> > > If you want empirical evidence of the computational practicality of our attack, please do check out our submitted code, where a FL simulation using the 3-layer Transformer can be attacked in a matter of seconds on a laptop CPU. If you wish to see even more references of FL for Transformer models, we have included an additional 25 references in a separate post below.
> > >
> > > &nbsp;
> > >
> > > >Second, why do we need "cross-silo" for keystroke prediction tasks? The benefit of FL is unclear here.
> > >
> > > There seems to be confusion over the scope of our attack. Our attack is not just for keystroke prediction tasks. In fact it is for any language modeling task which uses a Transformer-based architecture. Potentially motivating use-cases for hospitals include training a text suggestion model for doctors to speed up patient paperwork, or “In the context of electronic health records (EHR), for example, FL helps to represent and to find clinically similar patients” (see [2] from earlier). The benefit of FL here is that hospitals legally cannot share data, and so to train a model, a cross-silo FL setup is the only option.
> > >
> > > We strongly encourage you to browse the references we posted in our previous response and respond more specifically to them.
> > >
> > > &nbsp;
> > >
> > > >Second, a small amount of noise injected to the malicious parameters does not present state-of-the-art defenses against these types of attacks
> > >
> > > You seem to be confused about the two instances where we mention adding noise in our previous response.
> > >
> > > To clarify: the *first* time when we discuss adding noise, this is in the context of adding a “small amount of noise to the malicious parameters”. This is not a defense that *users* implement, but rather a defense that an *attacker* would implement to avoid a straightforward application of your suggested parameter inspection user defense. Speaking of which, do you have specific references to such a defense? What conditions would the defense specifically look for? Please see our previous response on how such a defense always puts users on the back foot.
> > >
> > > The *second* time we mention adding noise, is when we reference noise in “gradient clipping and noising in Section G of our Appendix (see Figure 18)”, which corresponds to a standard implementation of a differential privacy mechanism via DP-SGD.
> > >
> > > &nbsp;
> > >
> > >
> > > > This can be done by a trusted inspection system offered to the clients …
> > >
> > > Can you please reference such an inspection system? What conditions would the inspection system look for? Just the ones you mentioned originally? If so, please see our original response on why this is a bad idea as a defense, and is easily bypassed by the attacker.

---

> > > > ### Author Response · Authors · 2022-11-17
> > > > **References (2/2) our response**
> > > >
> > > > * X. Guo et al., “Federated Learning for Personalized Humor Recognition,” ACM Trans. Intell. Syst. Technol., vol. 13, no. 4, p. 68:1-68:18, May 2022, doi: 10.1145/3511710.
> > > >
> > > >
> > > > * H. Li et al., “FedTP: Federated Learning by Transformer Personalization.” arXiv, Nov. 02, 2022. doi: 10.48550/arXiv.2211.01572.
> > > >
> > > >
> > > > * S. Nalawade et al., “Federated Learning for Brain Tumor Segmentation Using MRI and Transformers,” in Brainlesion: Glioma, Multiple Sclerosis, Stroke and Traumatic Brain Injuries, Cham, 2022, pp. 444–454. doi: 10.1007/978-3-031-09002-8_39.
> > > >
> > > > * L. Qu et al., “Rethinking Architecture Design for Tackling Data Heterogeneity in Federated Learning,” presented at the Proceedings of the IEEE/CVF Conference on Computer Vision and Pattern Recognition, 2022, pp. 10061–10071. Accessed: Nov. 17, 2022. [Online]. Available: https://openaccess.thecvf.com/content/CVPR2022/html/Qu_Rethinking_Architecture_Design_for_Tackling_Data_Heterogeneity_in_Federated_Learning_CVPR_2022_paper.html
> > > >
> > > > * J. H. Ro et al., “Scaling Language Model Size in Cross-Device Federated Learning,” presented at the ACL 2022 Workshop on Federated Learning for Natural Language Processing, Mar. 2022. Accessed: Nov. 17, 2022. [Online]. Available: https://openreview.net/forum?id=ShNG29KGF-c
> > > > * M. Shaheen, M. S. Farooq, T. Umer, and B.-S. Kim, “Applications of Federated Learning; Taxonomy, Challenges, and Research Trends,” Electronics, vol. 11, no. 4, Art. no. 4, Jan. 2022, doi: 10.3390/electronics11040670.
> > > > * R. Sun et al., “FedMSA: A Model Selection and Adaptation System for Federated Learning,” Sensors, vol. 22, no. 19, Art. no. 19, Jan. 2022, doi: 10.3390/s22197244.
> > > > * J. Tao, Z. Gao, and Z. Guo, “Training Vision Transformers in Federated Learning with Limited Edge-Device Resources,” Electronics, vol. 11, no. 17, Art. no. 17, Jan. 2022, doi: 10.3390/electronics11172638.
> > > > * Y. Tian, Y. Wan, L. Lyu, D. Yao, H. Jin, and L. Sun, “FedBERT: When Federated Learning Meets Pre-training,” ACM Trans. Intell. Syst. Technol., vol. 13, no. 4, p. 66:1-66:26, Aug. 2022, doi: 10.1145/3510033.
> > > > * O. Weller, M. Marone, V. Braverman, D. Lawrie, and B. Van Durme, “Pretrained Models for Multilingual Federated Learning,” in Proceedings of the 2022 Conference of the North American Chapter of the Association for Computational Linguistics: Human Language Technologies, Seattle, United States, Jul. 2022, pp. 1413–1421. doi: 10.18653/v1/2022.naacl-main.101.
> > > > * B. Yuchen Lin et al., “FedNLP: Benchmarking Federated Learning Methods for Natural Language Processing Tasks,” Findings of NAACL, Jan. 2022, Accessed: Nov. 17, 2022. [Online]. Available: https://par.nsf.gov/biblio/10339512-fednlp-benchmarking-federated-learning-methods-natural-language-processing-tasks
> > > > * G. Zuccon, “Pretrained Language Models Rankers on Private Data: Is Online and Federated Learning the Solution?,” in 3rd International Conference on Design of Experimental Search & Information REtrieval Systems, San Jose, CA, USA, Aug. 2022, p. 5.

---

> > > > ### Author Response · Authors · 2022-11-17
> > > > **References (1/2) our response**
> > > >
> > > > * J. Chen, R. Zhang, J. Guo, Y. Fan, and X. Cheng, “FedMatch: Federated Learning Over Heterogeneous Question Answering Data,” in Proceedings of the 30th ACM International Conference on Information & Knowledge Management, New York, NY, USA, Oct. 2021, pp. 181–190. doi: 10.1145/3459637.3482345.
> > > > * R. Hathurusinghe, I. Nejadgholi, and M. Bolic, “A Privacy-Preserving Approach to Extraction of Personal Information through Automatic Annotation and Federated Learning.” arXiv, May 19, 2021. doi: 10.48550/arXiv.2105.09198.
> > > > * A. Hilmkil, S. Callh, M. Barbieri, L. R. Sütfeld, E. L. Zec, and O. Mogren, “Scaling Federated Learning for Fine-Tuning of Large Language Models,” in Natural Language Processing and Information Systems, Cham, 2021, pp. 15–23. doi: 10.1007/978-3-030-80599-9_2.
> > > > * Z. Hong, J. Wang, X. Qu, J. Liu, C. Zhao, and J. Xiao, “Federated Learning with Dynamic Transformer for Text to Speech,” in Interspeech 2021, Aug. 2021, pp. 3590–3594. doi: 10.21437/Interspeech.2021-2039.
> > > > * M. Liu, S. Ho, M. Wang, L. Gao, Y. Jin, and H. Zhang, “Federated Learning Meets Natural Language Processing: A Survey.” arXiv, Jul. 27, 2021. doi: 10.48550/arXiv.2107.12603.
> > > > * K. V. Sarma et al., “Federated learning improves site performance in multicenter deep learning without data sharing,” Journal of the American Medical Informatics Association, vol. 28, no. 6, pp. 1259–1264, Jun. 2021, doi: 10.1093/jamia/ocaa341.
> > > > * F. Sattler, T. Korjakow, R. Rischke, and W. Samek, “FEDAUX: Leveraging Unlabeled Auxiliary Data in Federated Learning,” IEEE Transactions on Neural Networks and Learning Systems, pp. 1–13, 2021, doi: 10.1109/TNNLS.2021.3129371.
> > > > * C. Thapa et al., “Evaluation of Federated Learning in Phishing Email Detection.” arXiv, May 21, 2021. doi: 10.48550/arXiv.2007.13300.
> > > > * C. Wang et al., “A Secure and Efficient Federated Learning Framework for NLP,” in Proceedings of the 2021 Conference on Empirical Methods in Natural Language Processing, Online and Punta Cana, Dominican Republic, Nov. 2021, pp. 7676–7682. doi: 10.18653/v1/2021.emnlp-main.606.
> > > > * A. Ait-Mlouk, S. A. Alawadi, S. Toor, and A. Hellander, “FedQAS: Privacy-Aware Machine Reading Comprehension with Federated Learning,” Applied Sciences, vol. 12, no. 6, Art. no. 6, Jan. 2022, doi: 10.3390/app12063130.
> > > > * E. Bagdasaryan, C. Song, R. van Dalen, M. Seigel, and Á. Cahill, “Training a Tokenizer for Free with Private Federated Learning.” arXiv, Mar. 15, 2022. doi: 10.48550/arXiv.2203.09943.
> > > > * P. Basu, T. S. Roy, R. Naidu, Z. Muftuoglu, S. Singh, and F. Mireshghallah, “Benchmarking Differential Privacy and Federated Learning for BERT Models.” arXiv, Jun. 16, 2022. doi: 10.48550/arXiv.2106.13973.
> > > > * Z. Charles, K. Bonawitz, S. Chiknavaryan, B. McMahan, and B. A. y Arcas, “Federated Select: A Primitive for Communication- and Memory-Efficient Federated Learning.” arXiv, Aug. 19, 2022. doi: 10.48550/arXiv.2208.09432.

---

> ### Author Response · Authors · 2022-11-12
> **Response (1/1)**
>
> Thank you for your time spent reviewing. Below we respond to the concerns you raise.
>
> &nbsp;
>
> >There is no evidence to support that clients train BERT or Transformers on their local devices for keystroke prediction tasks in real-world applications (deployment of such a setting)
>
> This criticism is mistaken for several reasons.
>
> 1. As we cite in the paper, the “3-layer Transformer” with which we experiment is taken directly from a highly cited field guide for federated optimization pioneered by Google (see Wang et al. 2021 in our manuscript). This is the same architecture that the authors of that work use for federated optimization for language models. In fact, the *only* architectures suggested for federated language tasks in the above work are Transformer-based.
>
> 2. Federated models can be trained on a variety of devices, including higher-powered devices, or devices that are always “plugged in” [1]. Especially in the “cross-silo” setting, federated optimization may actually take place on standard deep learning hardware, but because of regulations, e.g. medical privacy regulation, federated learning is employed to communicate model updates, for example, between hospitals [2]. Finally, devices (and batteries) will only continue to grow in capacity, and so too will the use-cases for federated learning which is still in a relatively nascent state. The use of larger Transformer models in the future seems all but guaranteed and security analysis in the present is the only way to remain pro-active against emerging threats.
>
> 3. The larger models with which we experiment also appear in *existing* literature for attacks against privacy in a federated language setting (see Zhu et al. 2019, Deng et al. 2021) and it is actually our inclusion of the small 3-layer model that is new - because we believe these smaller-scale models to be important. Furthermore, a litany of published and highly cited works on attacking privacy in federated learning use models even *larger* than some models on which we benchmark our attack. These include [3,4,5,6].
> Clearly, the architectures we consider are well within the realm of concern for the community at large.

---

> ### Author Response · Authors · 2022-11-17
> **Follow up**
>
> Are there any further questions we can address? We would be happy to discuss any remaining issues. If not, we would appreciate if you would consider raising your score based on our response.

---

> ### Author Response · Authors · 2022-11-23
> **Follow up (part 2)**
>
> We are politely following up about our last back-and-forth. Do you have any further questions?

---

### Official Review · Reviewer_7oyh · 2022-10-31

**Confidence:** 3
**Correctness:** 4
**Technical Novelty And Significance:** 3
**Empirical Novelty And Significance:** 4
**Recommendation:** 8

**Clarity, Quality, Novelty And Reproducibility:**

Clarity & Quality: The paper is presented in a fairly good flow. The work is solid.
Novelty: The setting/method is new in the NLP scope, although it is based on Fowl et al. (2021).
Reproducibility: The results should not be hard to reproduce.


**Strength And Weaknesses:**

Strengths:
- The paper proposes a novel attack method to reconstruct input text from gradients by carefully reprogramming the weights in transformer models. The attack method is interesting and makes sense to me.
- The paper is well-motivated and well-positioned. The attacking settings and threat models have been discussed well. The experimental results show the vulnerability of transformer models in a practical setting. I believe these results are insightful and are of interest to the NLP privacy/security community.
- The paper is presented in a good structure. I like the presentation in which a (simple) single-sentence case is presented first and then multi-sentence cases.

Weaknesses:
I don’t see a crucial weakness; but I would like to make the following suggestions which I think the paper can be improved upon.
- Figure 3 is not easy to follow, as long as the methods for reducing the probability of a collision.
- The authors are encouraged to provide more insights based on the experiments. For example, why is the 3-layer transformer more vulnerable?
- Is “total accuracy” a token-level metric or a sentence-level one?


**Summary Of The Paper:**

This paper focuses on the privacy problem under the setting of federated learning (FL), where a global model is trained by using the gradients computed by each private client. Although clients do not give out their private data but return the gradients computed using their data, the gradients may leak the private training data. The paper proposes an attack method that reconstructs the input text from the gradients. In this approach, the model weights will be reprogrammed such that by using the gradients, the attackers can solve the input text. Experimental results have shown that the proposed attack works well with different models, batch sizes, and sequence lengths.


**Summary Of The Review:**

In summary, I think this work is novel and solid. I hold a positive attitude towards accepting this paper.

---

> ### Author Response · Authors · 2022-11-12
> **Response**
>
> Thank you for your time and thoughtful comments. Below we respond to the concerns you raise.
>
> &nbsp;
>
> > Figure 3 is not easy to follow, as long as the methods for reducing the probability of a collision.
>
> Thank you for pointing this out. We have since modified the description of the process in the caption. Please let us know if there is still ambiguity.
>
>
> &nbsp;
>
>
> > The authors are encouraged to provide more insights based on the experiments. For example, why is the 3-layer transformer more vulnerable?
>
> In Figures 6,7, we actually see that, on average, the 3-layer Transformer is less vulnerable to the proposed attack. This is because the 3-layer Transformer has fewer parameters, and linear layers than the other two models (BERT-base and GPT-2). This means that, on average, fewer unique embeddings will be extracted by the attacker, and the recovered sequences will be of lower quality. We have added a comment about this in Section 4.
>
>
> &nbsp;
>
>
> > Is “total accuracy” a token-level metric or a sentence-level one?
>
> Total accuracy is simply the percentage of *exact* matches in the recovered sequences. An exact match occurs when the correct token is recovered *and* the exact correct position is recovered. So, for example, if the user sequence was “Hello, my name is Bob”, and the recovered sequence was “My name is Bob, hello”, then our total accuracy metric for this recovery would actually be 0%! Even though the attacker recovered all the correct words, each of the words’ positions is shifted. This makes this metric incredibly strict compared to metrics that other attacks use - for example just counting the number of n-grams recovered.

---

### Official Review · Reviewer_utwd · 2022-11-04

**Confidence:** 2
**Clarity, Quality, Novelty And Reproducibility:** This work is well structured and pres…
**Correctness:** 3
**Technical Novelty And Significance:** 2
**Empirical Novelty And Significance:** 3
**Recommendation:** 8

**Details Of Ethics Concerns:**

The approach will likely be used by malicious parties who can hijack the FL server.

**Strength And Weaknesses:**

**Strength:**
- The paper is well presented, where comprehensive background information is provided. Technical details are well documented and explained.
- One main contribution of this work is the threat model. As a conceptual contribution, the threat model is well-motivated and explained in detail. The attack scenario, as explained by the authors, is also interesting. However, there is also a concern regarding this threat model. Please check below.

**Weakness:**
- About the threat model contribution: will the malicious update be obvious to be detected by users themselves? After disabling all attention layers and most outputs of each feed-forward layer, will it be obvious for the users to realize the performance change of the transformer model? And, consequently, will it make the users stop adding new inputs? If so, this raises another relevant question: will the historical data be used to update the gradients?
- Given the different threat models, the results provided by Figure 9 are interesting but less informative, since the attack budgets are different in TAG and the proposed approach.



**Summary Of The Paper:**

This paper introduces an attack that corrupts the transformer models to extract private data from users' updates in federated learning. The threat model assumes that the server can send a malicious update to corrupt the user-side models and then can extract the original words and sentences entered from users' updates. To mount the attack, the adversary needs to be in charge of the server, send one malicious model update to the user side, and then extract private information.

**Summary Of The Review:**

This paper introduces an attack that corrupts the transformer models to extract private data from users' updates in federated learning. The threat model is novel and interesting. Comprehensive details are provided.

---

> ### Author Response · Authors · 2022-11-12
> **Response**
>
> Thank you for your time and energy in this review! Below we respond to the points you raise.
>
> &nbsp;
>
> > will the malicious update be obvious to be detected by users themselves?
>
> Please see General Comment 1 for our response regarding the difficulty for a basic user to distinguish models in early stages and malicious models.
>
> &nbsp;
>
> > will the historical data be used to update the gradients?
>
> If you are wondering about the user side - usually in federated learning, there is a “federated cache” (see Hard et al. 2019), which temporarily stores user data locally on the user’s device. Then, when the device is plugged in, and idle, federated training takes place on the data in the federated cache. The server receives updates computed on this cache, but the server has no requirement to include the compromised user update into their global model. The attack is ephemeral and can happen in a single round, while training of the global model proceeds normally in all other rounds. Let us know if you have further questions regarding this.
>
> &nbsp;
>
> > results provided by Figure 9 are interesting but less informative, since the attack budgets are different in TAG
>
> We agree that the threat model is different in TAG, which we note, and have clarified further. The included comparison is simply to show the orders-of-magnitude difference in vulnerability that our realistic threat model and attack induce for users. While we think the “honest but curious” threat model is an interesting setting (and a natural lower bound on attack success), ultimately we believe users should prepare for the worst-case scenario for privacy attacks.

---

### Official Review · Reviewer_yp9z · 2022-11-07

**Confidence:** 3
**Correctness:** 4
**Technical Novelty And Significance:** 3
**Empirical Novelty And Significance:** 3
**Recommendation:** 8

**Clarity, Quality, Novelty And Reproducibility:**

The paper is very clearly written, with a natural structure presenting the attack design before the recovery algorithm, and contains all necessary details. Notation is clear. Diagrams and illustrations are carefully executed and helpful. The paper is entirely self-contained so that only interesting but optional discussions are offloaded to the supplementary material.

The algorithm presented results from several, carefully assembled techniques (separating gradient signal, sequence reassignment, sequence vocabulary recovery, sequence clusters).

The experimental evaluation is careful, well described and informative in that it explores other threat models, attack confidence, factors affecting attack accuracy.

The paper is largely reproducible: code is provided with notebooks and a CLI; runs on CPU; uses very standard datasets.

This work is novel as it departs from previous, weaker threat models.

**Strength And Weaknesses:**

The paper is very strong and convinces through its clarity despite many algorithmic steps requiring the solution of non-trivial problems, as well as the strength of its results: it leaves no doubt that in the absence of any other security measure, federated learning, with models even as complicated as Transformers, widely allows recovering training sequences.

On principle, it does not come as a surprise that this recovery should be possible; but the practical demonstration of this conjecture, together with a careful analysis, was needed and is provided by this paper.

**Summary Of The Paper:**

The paper presents an attack on federated learning of Transformers which allows recovering the training sequences, based on a threat model in which the server is corrupt and submits a specially designed weight set which, once applied to the gradient computation of training sequences, will store all information necessary for the recovery of the training sequences in the gradient vectors.

**Summary Of The Review:**

The paper is of very good quality and established an important result, with consequences on future research and practical considerations.

---

> ### Author Response · Authors · 2022-11-12
> **Response**
>
> Thank you very much for your thoughtful review and time. We are glad you enjoyed our work. Please don’t hesitate to raise any further questions!

---

### Author Response · Authors · 2022-11-12
**General Comment**

We sincerely thank all the reviewers for their time and effort evaluating our submission. We are glad for the positive reception of our work!

Here, we address one common question, and we then respond to individual questions under their respective reviews.

&nbsp;

#### **1) On the visibility of the attack due to degraded model performance (Reviewers utwd, eNPF,  Fg6a)**

The models that are trained on user devices are usually not immediately deployed, and the production model that is deployed on devices is separate from the central model that is communicated during federated learning. For example, in Hard et al. 2019 (in our references), the authors only deploy the federated model for use on user devices *after* training, and only on a small proportion of experimental devices. This prevents failures during training from interrupting normal device interaction, and gives the server a chance to aggregate updates centrally before sending out a new production model (Appendix Section G).

Moreover, federated learning often takes place in the background, for example, at night when the device is charging (see Hard et al. 2019) and is a  regular occurrence on participating devices. Even if user applications were modified to make loss values on local data observable to the user, a user could not know whether a model was malicious, or whether a new model was simply in the early stages of training.

Finally, the attack can be quite ephemeral - the malicious server can send a *single* malicious update at the beginning of training, capture user data, and then send only benign updates from then on.

We’re happy to clarify this and we have included a brief explanation of this particular detail of FL
in the updated revision.

---

### Decision · Program_Chairs · 2023-01-20

**Decision:**

Accept: poster

**Justification For Why Not Higher Score:**

A better exploration of DP defenses was the main reason why a single reviewer maintained a very negative score.

**Justification For Why Not Lower Score:**

 Another reviewer chimed in and pointed out that attacks are of independent interest and have appeared in several papers in the past even without explicit defenses. All in all, even though a better exploration of DP defenses would add value to the paper, at the very least the paper has the potential of motivating exploration of exactly this topic. All other reviewers liked the paper.

**Metareview: Summary, Strengths And Weaknesses:**

The paper presents a novel federated learning attack, whereby a server sends weights to clients  appropriately designed so that it can recover the client inputs from client gradient computations. Several reviewers found it strong and convincing, particularly in (a) solving several constituent of non-trivial problems, (b) weakening assumptions/strengthening the attack compared to  previous work, (c) having a well motivated and interesting threat model and (c) establishing that even with models as complicated as Transformers, the attack allows recovering training sequences. The attack is tested on state of the art FL systems using transformers.

On the negative side, the most important criticism is that it is possible that the attack is severely hampered by the use of DP. Even though the paper includes an experimental study of DP defenses, this avenue is worth exploring further; seeing this explored more thoroughly in the paper would certainly improve the contribution, but discussions of attacks without countermeasures are also important contributions to the community-even for the purpose of exposing the inherent privacy issue, but also to further motivate DP research on the subject. Reviewers also would like to see more discussion and interpretation of certain phenomena observed in experiments, as well as a discussion on the visibility of the attack due to degraded model performance. The authors have provided responses to these issues; it would be good if they are also included in the paper.



**Note From Pc:**

if the above contains the word "oral" or "spotlight" please see: "oral" presentation means -> notable-top-5% and "spotlight" means -> notable-top-25%. As stated in our emails, we are disassociating presentation type from AC recommendations

**Summary Of Ac-Reviewer Meeting:**

This was not a borderline paper, and it was not discussed via zoom. There was a vigorous discussion with one of the reviewers online on several fronts, most of which the authors addressed, though the issue with regards to DP defenses remained. Another reviewer chimed in and pointed out that attacks are of independent interest and have appeared in several papers in the past even without explicit defenses. All in all, even though a better exploration of DP defenses would add value to the paper, at the very least the paper has the potential of motivating exploration of exactly this topic.